# Effect of Olive Pomace Oil on Cardiovascular Health and Associated Pathologies

**DOI:** 10.3390/nu14193927

**Published:** 2022-09-22

**Authors:** Susana González-Rámila, Beatriz Sarriá, Miguel Ángel Seguido, Joaquín García-Cordero, Laura Bravo-Clemente, Raquel Mateos

**Affiliations:** Department of Metabolism and Nutrition, Institute of Food Science, Technology and Nutrition (ICTAN-CSIC), Spanish National Research Council (CSIC), José Antonio Nováis 10, 28040 Madrid, Spain

**Keywords:** cardiovascular health, clinical trial, endothelial function, olive pomace oil, sunflower oil, visceral fat

## Abstract

Background: olive pomace oil (OPO) is a nutritionally relevant fat due to its high oleic acid content (C18:1) and the presence of a wide range of minor bioactive components. Although numerous in vitro and preclinical studies have been developed to study some of its characteristic components, the health effect of prolonged OPO consumption is unknown. Methods: a randomised, blinded, cross-over, controlled clinical trial was carried out in 31 normocholesterolemic and 37 hypercholesterolemic subjects. Participants consumed 45 g/day of OPO or sunflower oil (SO) for 4 weeks, each preceded by a 3-week run-in/wash-out phase with corn oil (CO). Results: regular consumption of OPO and SO had no statistically significant effect on any of the markers related to lipid profile, blood pressure, and endothelial function in both groups, except for eNOS levels, which were close to statistical significance due to the effect of oil (OPO and SO) (*p =* 0.083). A decrease in visceral fat (*p =* 0.028) in both groups was observed after OPO intake, accompanied by an increment of leptin (*p =* 0.017) in the hypercholesterolemic group. Conclusion: reducing visceral fat after prolonged OPO intake might contribute to improve cardiometabolic status, with a potentially positive effect on the vascular tone. Further clinical trials are needed to confirm the present results.

## 1. Introduction

Cardiovascular disease is the prevalent cause of morbidity and mortality worldwide, affecting millions of individuals every year. One of the most relevant environmental factors contributing to the development of this pathology is the diet [1]. In this sense, the Mediterranean Diet (MD) is considered one of the best models of healthy eating due to its beneficial effects on chronic non-communicable diseases, such as cardiovascular disorders, diabetes, obesity, and inflammation [2]. The main source of fat in this dietary pattern is olive oil [2]. This oil is obtained exclusively from olives and, depending on the technological process to which it is subjected, different commercial categories are obtained: extra virgin olive oil (EVOO), virgin olive oil (VOO), olive oil (OO), mild and intense), and olive pomace oil (OPO) [3]. The health benefits attributed to olive oil consumption, particularly reducing the risk of cardiovascular diseases, have been related to EVOO and VOO consumption due to its preferably monounsaturated fat and its high content of phenolic compounds [4,5]. However, in countries such as Spain, where the olive oil consumption is deeply rooted, the main competitors in the olive market are seed oils (high oleic sunflower oil (HOSO), sunflower oil (SO), etc.) [6]. These oils are mainly used for frying, and their low economic costs are the main reason for their elevated consumption [7,8]. This situation represents an opportunity for OPO because its composition gives the ideal characteristics for frying and, in addition, its market value is quite competitive [9,10].

OPO is obtained from alperujo, a solid by-product consisting of pieces of olive skin, pulp, pits, and stones [3,11]. This oil is characterised by its high oleic acid content, and its specific composition in minor components, mainly triterpenic acids and dialcohols, squalene, tocopherols, sterols, aliphatic fatty alcohols, and phenolic compounds [3,9]. The refining process carried out on this oil causes the loss of some of the triterpenic acids and phenolic compounds contained in alperujo [3]. However, concentrations of the minor components present in the refined pomace oil are found in amounts that are likely to induce beneficial effects on health, particularly on cardiovascular health [3].

In fact, numerous in vitro and preclinical studies have evaluated one or more components of OPO’s minor fraction with promising results [12,13,14,15]. In addition, the long-term effect of OPO consumption at nutritional doses has been evaluated in two clinical trials developed by our research group. In the first clinical trial, OPO was compared to HOSO [16], whereas in the second, OPO was compared to SO, a polyunsaturated fat. This present study is focused on the second study. Recently, the comparison of the two clinical trials has been published [17].

Thus, the aim of this study was to determine the possible beneficial role of OPO on biomarkers of cardiovascular health and associated pathologies (hypertension, inflammation, diabetes, and obesity) in healthy volunteers and subjects at cardiovascular risk (hypercholesterolemic volunteers), in order to comparatively test the dietary treatment in both groups. The effect of OPO was compared to SO (control fat), given the high consumption of this seed oil. Corn oil (CO), a seed oil other than SO, was used as run in/wash out oil.

## 2. Materials and Methods

### 2.1. Study Design

The study was a randomized, blind, crossover, and controlled clinical trial with a duration of 14 weeks. After a 3-week run-in stage, in which all volunteers consumed CO to wash out the effect of oil consumed in their habitual diet, half of the participants were randomly assigned to the OPO and the other half to the SO oil, both in the healthy and in the at-risk groups. After the first 4-week intervention, a 3-week wash-out period with CO followed, and subsequently, participants consumed the other oil (OPO or SO) during the same period of 4 weeks (Figure 1). Randomization of participants to OPO or SO (control) interventions was performed using Microsoft^®^ Excel 2016 software in a 1:1 ratio. Assignment of codes to participants, randomization and allocation to each oil were carried out by different members of the research team. To blind participants, oils were presented in identical plastic bottles with different taps for each type of oil (OPO, SO, and CO).

### 2.2. Participants and Setting

Of the 109 subjects interviewed, the volunteers who met the inclusion criteria (age between 18–55 years, body mass index (BMI) between 18–25 kg/m^2^; specific inclusion criteria for healthy and at-risk volunteers were based on serum lipid levels as stated below), were 43 women and 29 men, aged 19 to 59 years old, with a BMI between 19.0 and 32.0 kg/m^2^. The study groups were formed according to the serum total and LDL-cholesterol (LDL-C) concentrations determined in a pre-screen consented blood analysis. Healthy participants were assigned to the normocholesterolemic group when total cholesterol (TC) and LDL-C levels were below 200 mg/dL and 135 mg/dL, respectively. At-risk volunteers were assigned to the hypercholesterolemic group when TC was between 200–300 mg/dL and LDL-C between 135–175 mg/dL (Figure 2). The exclusion criteria established were: suffering from acute or chronic pathologies, except hypercholesterolemia for the risk group, having digestive disorders/pathologies (gastric ulcer, Crohn’s disease, inflammatory bowel syndrome, etc.), smoking, pregnant women, vegetarians, on antibiotic treatment three months before starting the study, or taking medication, hormones, or vitamins or dietary supplements.

The recruitment was conducted between June and September 2019 mainly through our database of participants in previous studies, social networks and by placing flyers at the Institute of Food Science, Technology and Nutrition (ICTAN) and at the Complutense University of Madrid campus (UCM). Volunteers interested in the study received additional information by e-mail and telephone. Finally, those who met the inclusion criteria were invited to a meeting to explain the study in detail, and answer questions raised by the attendees. The study was approved by the Clinical Research Ethics Committee of the Hospital Universitario Puerta de Hierro, at Majadahonda in Madrid (Spain), and by the Bioethics Committee of the Consejo Superior de Investigaciones Científicas (CSIC). It also followed the guidelines laid down in the Declaration of Helsinki for experiments in humans. Before starting the study, we ensured that we obtained signed informed consent from all participants. The study was registered in Clinical Trials (NCT04998695).

### 2.3. Intervention

The intervention study was conducted during October-December 2019 at the Human Nutrition Unit (HNU) of the ICTAN. All volunteers consumed 45 g/day of oil according to the intervention phase (OPO, SO, and CO) to cover 20% of the daily energy intake of monounsaturated fats (equivalent to 44–67 g/day for 2000–3000 Kcal/d, respectively) following the Spanish Society of Community Nutrition (SENC) recommendations. To this end, one litre of oil per week was provided for family consumption to avoid using any other culinary oil. Participants were required to maintain their dietary and lifestyle habits unchanged, except for some foods rich in mono- and polyunsaturated fat (olives, sunflower seeds, nuts, avocado, margarine, butter, and mayonnaise, except if prepared with the study oils), which were restricted during the study.

### 2.4. Chemical Characterization of the Study Oils

The oils used in the study were analysed according to the following standardized methods:

ISO 12228-2:2014 method for the determination of sterols.

Regulation (EEC) N°. 2568/91 Annex V for determining triterpenic alcohols.

Regulation (EEC) N°. 2568/91 Annex XIX for determining aliphatic alcohols.

Regulation (EEC) N°. 2568/91 Annex X for determining fatty acid composition.

ISO 9936:2016 for determining tocopherols and tocotrienols.

Triterpenic acids were analysed following the method of Pérez-Camino & Cert [18], and squalene was determined by gas chromatography [19]. Phenols were analysed by high-performance liquid chromatography with on-line diode array detection (HPLC-DAD) according to the procedure developed by Mateos et al. [20].

### 2.5. Dietary Assessment and Compliance

As indicated in the intervention section, participants were asked to maintain their dietary habits and lifestyle unchanged during the study. To assess if they did, at each visit, participants completed a 24-h questionnaire for the day prior to the visit, and a detailed record of food intake for 72 h, in which they recorded the ingredients, units, or quantity of food, as well as the time and place where they ate. A Manual of Nutrition and Dietetics [21] was provided to the participants to facilitate the interpretation of home food measurements and usual portions. Compliance with food restrictions and the correct intake of oil was controlled by weekly calling and/or emailing participants.

Macronutrients (proteins, carbohydrates, and lipids), dietary fibre and vitamin E intake were calculated using the program DIAL (Department of Nutrition and Bromatology, Faculty of Pharmacy, Complutense University of Madrid, Madrid, Spain), as well as to estimate the total caloric intake.

### 2.6. Blood Sample Collection

Fasting blood samples were collected after an overnight fast using BD Vaccuette^®^ tubes (Greiner Bio-One GmbH, Kirchdorf an der Krems, Austria) with EDTA or without anticoagulant to separate plasma and serum, respectively. Subsequently, samples were centrifuged, aliquoted, and stored at −80 °C until analysis.

### 2.7. Primary Outcomes and Other Outcomes Measures

Primary outcomes were markers related to cardiovascular health: lipid profile, blood pressure, and endothelial function. Secondary outcomes were biomarkers associated with diabetes, obesity, and inflammation.

#### 2.7.1. Biochemical Analysis

Biochemical analysis of metabolic markers in serum was performed following procedures of reference as established by the Spanish Society of Clinical Biochemistry and Molecular Pathology (Sociedad Española de Bioquímica Clínica y Patología Molecular (SEQC). TC, triglycerides (TG), high-density lipoprotein (HDL) cholesterol, apolipoprotein A1 (Apo A1), and apolipoprotein B (Apo B) were determined using a Roche Cobas Integra 400 plus analyser (Roche Diagnostics, Mannheim, Germany). In addition, the Friedewald formula was used to estimate LDL-C and VLDL (very low-density lipoprotein) content, and Apo B/Apo A1, LDL/HDL, and TC/HDL ratios were calculated. Alanine (ALAT) and aspartate (ASAT) aminotransferases were analysed spectrophotometrically following standard procedures.

#### 2.7.2. Blood Pressure

Systolic blood pressure (SBP) and diastolic blood pressure (DBP) were measured with an OMRON^®^ M2 HEM-7121-E sphygmomanometer (OMRON HEALTHCARE Co., Ltd., Kyoto, Japan). Participants rested in a sitting position for 15 min before measurement, which was taken in triplicate in the non-prevailing arm, waiting 5 min between each.

#### 2.7.3. Endothelial Function

Circulating levels of E-selectin, P-selectin, and the enzyme endothelial nitric oxide synthase (eNOS) were determined in plasma by ELISA following the protocols of Cloud-Clone Corp. (Katy, TX, USA). Spectrophotometric reading was performed using a Bio-Tek^®^ Synergy™ HT Multi-Detection microplate reader controlled by BioTek^®^Gen5 software version 2.01.14 (BioTek Instruments, Winooski, VT, USA). Intercellular (ICAM-1) and vascular (VCAM-1) adhesion molecules were determined in serum samples with Bio-Plex^®^ Pro Human Cytokine ICAM-1 and VCAM-1 kits (Bio-Rad, Hercules, CA, USA) using a MAGPIX™ Multiplex fluorescence reader operating with the Bio-Plex Pro Wash Station and the Bio-Plex Manager™ MP software for data processing (Luminex Corporation, Austin, TX, USA).

#### 2.7.4. Diabetes and Obesity Biomarkers Analysis

Glucose, insulin, and glycosylated haemoglobin (HbA1c) were determined in serum samples following recommendations of the Spanish Society of Clinical Biochemistry and Molecular Pathology (SEQC). From the glucose and insulin data, insulin resistance (HOMA-IR) and pancreatic beta-cell function (HOMA-β) were calculated according to the mathematical model known as HOMA (Homeostasis Model Assessment) proposed by Matthews et al. [22]: HOMA-IR = [Glucose (mg/dL) × Insulin (mU/L)]/405; HOMA-β = [(360 × Insulin (mU/L)/(Glucose (mg/dL) − 63)]. In addition, insulin sensitivity was determined using the QUICKI (“Quantitative Insulin Sensitivity Check Index”), based on a logarithmic model calculated from fasting glucose and insulin concentrations using the following equation QUICKI = 1/[log Insulin (mU/L) + log Glucose (mg/dL)].

The hormones insulin, glucagon, incretin gastric inhibitory polypeptide (GIP), as well as glucagon-like peptide type 1 (GLP-1), C-peptide, and ghrelin; along with the adipokines leptin, resistin, plasminogen activator inhibitor-1 (PAI-1), visfatin, adiponectin, and adipsin were analysed in serum samples using the Bio-Plex^®^ Pro^TM^ Human Diabetes Panel 10-Plex kit (Bio-Rad, Hercules, CA, USA) in the MAGPIX™ Multiplex fluorescence reader and Bio-Plex Manager™ MP software (Luminex Corporation, Austin, TX, USA).

#### 2.7.5. Anthropometry and Body Composition

Height and body perimeters (waist, abdomen, and brachial) were measured in triplicate using a wall-mounted height measuring rod (Soehnle Professional, GmBH, Backnang, Germany) and a tape measure (Fisaude ADE, Madrid, Spain), respectively. Weight, visceral, and body fat percentages were estimated by single-frequency tetrapolar electrical bioimpedance using a Tanita^®^ BC 601 segmental body composition analyser with a digital scale included (Tanita Europe BV, Amsterdam, The Netherlands).

#### 2.7.6. Inflammatory Biomarker Analysis

Pro-inflammatory (IL-1β, IL-2, IL-6, IL-7, IL-8, IL-12(p70), IL-17) and anti-inflammatory (IL-4, IL-10 and IL-13) interleukins (IL), interferon-gamma (IFN-γ), and tumour necrosis factor alpha (TNF-α), as well as monocyte chemoattractant protein 1 (MCP-1), macrophage inflammatory protein 1 beta (MIP-1β), granulocyte colony-stimulating factor (G-CSF), and granulocyte monocyte colony-stimulating factor (GM-CSF) were analysed in serum samples using Bio-Plex Pro^TM^ Human Cytokine Grp I Panel 17-Plex kits in the MAGPIX™ Multiplex fluorescence reader and Bio-Plex Manager™ MP software (Luminex Corporation, Austin, TX, USA). In addition, high-sensitivity C-reactive protein (CRP) was also determined in serum samples with an automated ultra-sensitive turbidimetric method (AU2700 Chemistry Analyzer, Olympus Corp., Japan).

#### 2.7.7. Antioxidant Capacity and Oxidation Biomarkers

Antioxidant activity was measured in serum samples by the ABTS radical cation [23] and the oxygen radical absorbance capacity (ORAC) methods [24], and the reducing capacity was determined by the ferric reducing/antioxidant power (FRAP) assay [25]. Trolox was used as standard, and results were expressed as μM of Trolox equivalent (TE). Low-density lipoprotein oxidation (LDLox) levels were determined in serum samples by ELISA assay according to the protocols of the Cloud-Clone Corp. kit (Katy, TX, USA). These parameters were analysed using a Bio-Tek^®^ Synergy™ HT Multi-Detection plate reader (Highland Park, Winooski, Vermont 05404-0998 USA) controlled by BioTek^®^Gen5 software version 2.01.14.

Malondialdehyde (MDA) levels, a biomarker of lipid oxidation, was determined in serum samples by high-performance liquid chromatography (HPLC) following the methodology proposed by Mateos et al. [26]. For this purpose, a 1200 series HPLC equipment (Agilent Technologies, Santa Clara, CA, USA) and a Nucleosil 120 C18 column (25 mm × 0.46 mm, particle size 5 µm, TeknoKroma, Barcelona, Spain) were used.

### 2.8. Sample Size Calculation and Statistical Analysis

To estimate the sample size, the G*Power 3.1.9.7 program was used, considering TC concentration as the main variable and the study design [randomized, blind, crossover, and controlled clinical trial, in which all subjects consumed both test (OPO) and control oils (SO)]. Other premises considered, following previous studies with a similar design, were: a statistical power of 80%, a level of significance of 0.05, two tail, a standard deviation of 25, mean of pre-post differences of 13.4 units, and an effect size of 0.54 [27]. A sample size of 30 volunteers was established. Finally, considering their TC and LDL-C concentration, 72 subjects were recruited and allocated in the normocholesterolemic or hypercholesterolemic groups, although only *n* = 31 healthy and *n* = 37 at risk volunteers completed the study. These numbers were higher than the sample size of 30 initially calculated and allowed for adequate statistical analysis.

For the statistical design, the following factors were considered as two fixed effects: group (normocholesterolemic/hypercholesterolemic) and treatment (OPO/SO, repeated measures), and the order of oil intake (starting with OPO or SO within each group) was considered as a random effect.

The statistical models applied to analyse the results of this study were:

1. A general linear repeated measures model to study energy, macronutrient, and micronutrient intake throughout the study, considering that the order of intake of the test and control oils would not affect the overall dietary pattern of the volunteers. In each group (normocholesterolemic and hypercholesterolemic), baseline, initial (pre-treatment), and final (post-treatment) results with OPO and SO were compared. Results are shown as mean ± standard error of the mean (SEM).

2. A linear mixed model was applied to study the rate of change [(final value—initial value)/initial value] of each variable. This statistical model considers the order of intake of the oils, presenting the data in a correlated and non-constant variability form. This statistical model was also applied to the initial and final mean values. The statistical model was full factorial, considering that group (normocholesterolemic hypercholesterolemic), treatment oil (OPO and SO) and interaction group*treatment. Pre- and post-treatment data are shown as mean ± standard error of the mean (SEM), and the rate of change is expressed as a percentage ± SEM.

Normality of data distribution was verified by the Kolmogorov-Smirnov test, and a box-plot analysis was performed for all variables before statistical analysis. In addition, the Bonferroni test (within each group) was applied to compare pairwise the effect of the intake of each oil (OPO and SO). The significance level was set at *p <* 0.05. Data were analysed using SPSS software (version 27.0; SPSS, Inc., IBM Company, Armonk, NY, USA).

## 3. Results

### 3.1. Chemical Composition of the Study Oils

The chemical characterisation of OPO, SO, and CO is shown in Table 1. OPO was a monounsaturated fat with an oleic acid (C18:1) content of 74.32%, followed by palmitic acid (C16:0), and linoleic acid (C18:2) with a content of 10.78% and 9.02%, respectively. Linoleic acid (C18:2) was the main fatty acid in SO (58.46%) and in corn oil (CO) (50.52%), followed by oleic acid (C18:1), with a content of 29.76% in SO and 35.36% in CO. As for minor components, squalene was present in the three oils, with a higher concentration in OPO (675 ppm) compared to SO (314 ppm) and CO (548 ppm). In the case of tocopherols, while OPO (195 mg/kg) and SO (217 mg/kg) showed a high content of α-tocopherol (vitamin E), CO stood out for its γ-tocopherol content (205 mg/kg). Tocotrienols were only detected in CO (19 mg/kg). Regarding sterols composition, the content was highest in CO (8962 ppm), followed by OPO (3344.2 ppm) and SO (2820.5 ppm). However, it should be noted that triterpenic alcohols (745 mg/kg) and triterpenic acids (191 mg/kg), as well as aliphatic alcohols (1681 mg/kg), were mainly detected in OPO, compared to their low or no content in SO and CO, respectively. Finally, the phenol content was below 2 mg/kg in the three oils (Table 1).

### 3.2. Baseline Characteristics of Participants and Dietary Control

Of the 109 screened for assessment, only 72 participants were recruited. Two participants dropped out due to incompatibility with their work, one due to medical prescription, and one was lost to follow-up (the study flow diagram is shown in Figure 2). Thus, 68 participants (31 normocholesterolemic and 37 hypercholesterolemic) successfully completed the study. Although an effort was made to ensure equal representation of both sexes in the two groups, at the end there were more normocholesterolemic women (23) than men (8), although hypercholesterolemic men (19) and women (18) were balanced. The baseline characteristics of participants are presented in Table 2. Two hypercholesterolemic volunteers presented unusually high inflammatory biomarkers (outliers) and thus were excluded in the statistical analysis of inflammation results.

Table 3 shows energy, macronutrient, micronutrient, and dietary fibre intake during the study. Energy, protein, carbohydrate, and lipid values did not change significantly in both population groups after OPO and SO intervention (*p <* 0.05) (Table 3). These data confirm that the volunteers followed the instructions of not changing their dietary habits during the study. The total caloric intake showed mean values between 1906 and 2081 Kcal/day, slightly below normal limits [28]. In relation to macronutrient intake, the mean values for proteins (84 g/day), carbohydrates (188 g/day), and lipids (92 g/day) represented 16.7%, 37.5%, and 41.3%, respectively. In accordance with the recommended daily intake and nutritional targets for the Spanish population, the data for proteins and lipids were found to be above the recommendations and, in the case of carbohydrates, their values were under the recommendation (50–60% of the total diet) (Table 3) [28].

Saturated fatty acids (SFA) intake values ranged between 28 and 31 g/day (12.6 and 14% of the total diet), above 7–8% of the recommended total daily energy [28]. However, OPO and SO consumption did not show significant variations in SFA values (*p* < 0.05). As for monounsaturated fatty acids (MUFA), significant changes (*p* < 0.001) were observed in both groups after OPO and SO intake. According to the multiple comparisons test, the results showed that, after regular consumption of OPO, there was a significantly increased consumption of this nutrient (with a mean value of 44.5 g/day), reaching 20% of the total energy of the diet [28]. In contrast, MUFA values remained constant after the SO intervention, with an average intake of 29.5 g/day, which represented 13.3% of the total energy [28] (Table 3). This result was expected considering that OPO is a rich source of MUFA compared to SO, whose fat is mostly polyunsaturated (Table 1). Polyunsaturated fatty acid (PUFA) intake showed significant changes (*p* < 0.001) in the normocholesterolemic and hypercholesterolemic groups after the intervention with OPO and SO. The Bonferroni test revealed that, in line with the dietary intervention, after OPO consumption, there was a significant decrease in PUFA intake (11–13 g/day, equivalent to 5–6% of total dietary energy), and an increase after regular intake of SO (29–31 g/day, equivalent to 13–14% of total dietary energy). It is worth mentioning that in the run-in and wash-out stages (before the start with OPO and SO intervention), the mean PUFA values (22–27 g/day, equivalent to 10–12%, respectively) were higher than those observed with basal diet consumption (13.5 g/day, equivalent to 6% of total energy). The reason was that the CO used during these stages (run-in and wash-out) is a rich source of PUFA. The recommendation of 5% PUFA of total dietary energy was close to those achieved after the OPO intervention (5–6% of total dietary energy) and the basal stage (6% of total energy) (Table 3).

### 3.3. Blood Pressure

No significant differences were found in SBP and DBP after the dietary intervention with OPO and SO. However, there were significant differences in SBP and DBP between the normocholesterolemic and hypercholesterolemic groups, when initial and final values were analysed (*p* < 0.05) (Appendix A). According to the European Society of Cardiology (ESC), all participants maintained blood pressure values within the normal range of <120 mmHg for SBP and <80 mmHg for DBP

### 3.4. Blood Biochemistry: Lipid Profile and Liver Function

As shown in Table 4, the volunteers’ lipid profiles did not change (*p* < 0.05) after prolonged consumption of OPO and SO in either group (normocholesterolemic and hypercholesterolemic). However, the trend towards a decrease in circulating levels of TC, TG, LDL-C, and VLDL-C after OPO intervention in the healthy group is noteworthy. On the other hand, when the linear mixed model was applied to the initial and final values, TC, TG, LDL-C, VLDL-C, LDL-C/HDL-C, TC/HDL-C, Apo A1, Apo B, and Apo B/Apo A1 ratio showed differences between the normocholesterolemic and hypercholesterolemic groups (*p* < 0.05). In relation to the ALAT and ASAT enzymes, there were some differences between the groups at the initial and end of the intervention, with higher values (*p* < 0.05) in the hypercholesterolemic subjects. When the rate of change was analysed, ASAT enzyme also showed significantly higher values in the hypercholesterolemic group compared to the healthy group (*p* = 0.045). However, none of the studied oils had a significant effect on these enzymes.

### 3.5. Inflammatory Biomarkers

After OPO and SO intervention, statistically significant changes were observed in IL-4 (*p* = 0.021) and IL-13 (*p* = 0.023). According to Table 5, IL-4 showed important changes by the interaction of the normo- and hypercholesterolemic groups with the oils (OPO or SO), with an increase and a decrease after OPO and SO intake, respectively. With respect to IL-13 (*p* = 0.023), there were significant changes after prolonged consumption of both vegetable oils, with an increase after the OPO intake, and a decrease after the SO consumption (Table 5).

### 3.6. Biomarkers of Endothelial Function

Endothelial function biomarkers (Table 6) showed no significant changes throughout the study according to the linear mixed model applied on the rates of change (*p* < 0.05). However, it is noteworthy that circulating eNOS concentrations were close to the level of significance (set at *p* < 0.05) in both groups, with an increasing and decreasing trend after OPO and SO intake, respectively.

### 3.7. Diabetes Markers: Glycaemia, Insulin Levels, Glycosylated Haemoglobin Concentration, Insulin Resistance/Sensitivity Indices (HOMA-IR/QUICKI), and Pancreatic Beta-Cell Function (HOMA-β)

To assess insulin resistance and glycaemic homeostasis, mean glucose, insulin, and HbA1c values were analysed. As shown in Table 7, glucose (*p* = 0.007) showed significant differences after OPO and SO dietary intervention due to the interaction of the group (normo- and hypercholesterolemic) with the studied oils (OPO and SO). According to the Bonferroni test, there were differences between OPO and SO in subjects with normal cholesterol levels. Thus, glucose values increased by 1.3% after OPO intake and decreased by 2.6% after SO intake. In the case of insulin, there were no statistically significant variations throughout the study, although it was close to being significant due to the effect of consuming OPO and SO (*p* = 0.077). Regarding Hb1Ac (*p* = 0.021), the differences occurred between groups, with slightly higher values in the hypercholesterolemic group. Importantly, despite the observed changes in glucose and Hb1Ac, the values of both parameters were within normal levels for the adult population (<6% and <120 mg/dL, respectively) established by the Spanish Diabetes Federation (Federación Española de Diabetes, FEDE). The mathematical models proposed by Matthews et al. [22] to assess insulin resistance (HOMA-IR) and pancreatic beta-cell function (HOMA-β), as well as the QUICKI index calculated for insulin sensitivity, did not show statistically significant differences after consumption of OPO and SO (*p* < 0.05). However, differences were observed between healthy and at-risk subjects in HOMA-IR (*p* = 0.020) and QUICKI (*p* = 0.011) values. Finally, after linear mixed model analysis of initial and final values, HOMA-IR (*p* = 0.006) and QUICKI index (*p* = 0.005) revealed differences between normocholesterolemic and hypercholesterolemic subjects.

### 3.8. Obesity Biomarkers

To determine the possible beneficial role of OPO and SO consumption on diabetes and obesity, several biomarkers closely related to these two pathologies were analysed (Table 8). According to the linear mixed model applied to rates of change, the dietary intervention with OPO and SO showed significant changes in the hormones ghrelin (*p* = 0.031) and leptin (*p* = 0.017). Specifically, ghrelin values increased in both groups after OPO and SO intervention, being more marked in cardiovascular risk subjects after SO intake. Regarding leptin, values in healthy subjects decreased by −8.0% after OPO intervention and slightly increased by 1.7% after SO intervention, while there was an increase of 8.6% and 18.6% after OPO and SO intake, respectively, in at-risk subjects. All other variables analysed showed no changes. In addition, belonging to the normocholesterolemic or hypercholesterolemic group had significant effects on C-peptide (initial and final stages), insulin (final stages), and visfatin (final stages) when initial and final values were analysed (*p* < 0.05). On the other hand, adiponectin showed changes due to the effect of the oils (OPO and SO) when final values were compared (*p* = 0.029).

### 3.9. Anthropometric Measurements and Body Composition

Table 9 shows how 4-week consumption of OPO and SO significantly affected visceral fat percentage (*p* = 0.028). In detail, there was a decrease of −4.5% (normocholesterolemic group) and −1.4% (hypercholesterolemic group) after OPO intake, and an increase of 2.2% (normocholesterolemic group) and 6.6% (hypercholesterolemic group) after SO intake. The other parameters assessed were not affected after the intake of both vegetable oils (OPO and SO). However, when healthy subjects were compared with those at-risk, differences in BMI (*p* = 0.050) and body fat (*p* = 0.058) values were observed. These differences between groups were also observed in weight (initial and final stages), BMI (initial and final stages), visceral fat (initial and final stages), waist circumference (initial and final stages), hip circumference (late stages), and arm circumference (initial and final stages) (*p* < 0.05), when the statistical analysis was performed on initial and final values (Table 9).

### 3.10. Antioxidant Capacity and Oxidation Biomarkers

The consumption of OPO and SO had no effect on any of the parameters used to assess the oxidative status of the participants (*p* < 0.05). However, belonging to the normocholesterolemic or hypercholesterolemic group influenced FRAP values (*p* = 0.013). In addition, initial and final FRAP and LDLox values showed variations between the normocholesterolemic and hypercholesterolemic groups, with slightly higher values in the at-risk group (*p* < 0.05) (Appendix A).

## 4. Discussion

The MD is one of the best-studied diets because of its beneficial, protective effect against chronic and inflammatory diseases. One of the most well-known and important characteristics of this diet is the use of olive oil as the principal source of energy from fat [2]. All varieties of olive oil (EVOO, VOO, OO, and OPO) are characterised by their high oleic acid content and their wide variety of bioactive compounds, while chemical composition depends on the extraction method [3]. OPO is obtained from the residue remaining after mechanical extraction of the VOO [9]. Although it is refined to make it suitable for consumption, it remains a rich source of triterpenic acids and dialcohols, squalene, tocopherols, sterols, and aliphatic fatty alcohols [3]. Numerous in vitro and preclinical studies have been conducted on some of the minor components of OPO showing promising results in the prevention of cardiovascular diseases and their risk factors, modulating lipid profile, improving endothelial function, inducing hypotensive effects, lowering inflammation levels, and improving biomarkers related to the prevention of diabetes and obesity [3]. However, the possible effects on human health following prolonged OPO consumption remain unknown. Therefore, a randomized, blinded, crossover, controlled clinical trial with OPO, analysing numerous biomarkers related to heart disease and associated pathologies (obesity, diabetes, and inflammation) has been carried out. 

### 4.1. Chemical Composition of the Study Oils

OPO is mainly a monounsaturated fat thanks to its high oleic acid (C18:1) content in comparison with SO and CO, and it is richer in linoleic acid (C18:2). These results were in line with those recently reported by Holgado et al. [29], who characterized three different brands of OPO and SO. The oleic acid (72.02–73.8% for OPO; 28.38–32.16% for SO) and linoleic acid (9.5–11.1% for OPO; 56.6–60.0% for SO) content was similar to those quantified in our study oils (Table 1). These authors also analysed other minor components, with some differences between the OPO used in this study and those characterized by Holgado et al. [29]. The three different OPO analysed by these authors showed a higher content of squalene (742–1538 mg/kg), tocopherols (301–446 mg/kg), and phenolic compounds (8–15 mg/kg), and a lower amount of aliphatic fatty alcohols (1677–2269 mg/kg) than the OPO used in this intervention study (squalene: 675 mg/kg; tocopherols: 201 mg/kg; phenolic compounds < 2 mg/kg; aliphatic fatty alcohols: 1681 mg/kg, Table 1). Regarding CO composition, oleic acid, linoleic acid, and total sterol content shower a similar level to that reported by Güneşer et al. [30].

### 4.2. Effect of Nutritional Intervention on Lipid Profile, Blood Pressure, and Markers of Liver Function and Endothelial Function

Prolonged consumption of OPO and SO did not induce changes (*p <* 0.05) in any of the biomarkers analysed related to lipid profile and liver function (Table 4). Although the hypolipidemic effect of OPO had not been tested before, numerous in vivo and in vitro studies performed with OPO components (mainly triterpenic acids and aliphatic fatty alcohols) showed a favourable influence on various lipid profile parameters [3]. In addition, oleic acid, as the main fatty acid in all categories of olive oil (EVOO, VOO, OO, and OPO), allows these oils to be classified as monounsaturated fats with the potential to reduce coronary heart disease risk, as had already been recognized by the FDA (Food and Drug Administration 2004) [31]. In this regard, a systematic review and meta-analysis developed by Pastor et al. [32] to assess the effect of hydroxytyrosol, oleic acid, or a combination of both (olive oil) on metabolic syndrome revealed that oleic acid consumption had a beneficial effect on lipid profile. Therefore, the slight downward trend observed in serum levels of TC (in both groups), and TG, LDL-C, and VLDL-C in healthy subjects after OPO intake (Table 4), yet not reaching significant differences, could be a consequence of consuming this monounsaturated fat (OPO) rich in bioactive compounds. In relation to SO, its effect on the lipid profile was investigated in 200 patients with heart disease, where consumption of SO versus coconut oil (a source of saturated fat) showed no significant differences in lipid profile markers (TC, LDL-C, HDL-C, VLDL, Apo B/Apo A ratio) [33], in agreement with the results obtained in the present study. As shown in Table 4, the intervention with SO also resulted in slight, non-significant decreases in TG and VLDL levels in healthy subjects and a reduction in TC in the risk group. This downward trend may be related to the high linoleic acid content of SO, since consumption of this PUFA has been associated with reduced CVD mortality [34].

On the other hand, the mean values of ASAT and ALAT enzymes did not show variations in both groups (normocholesterolemic and hypercholesterolemic) after OPO and SO intake (Table 4). This result was to be expected considering that an increase in the concentration and/or activity of these enzymes is related to adverse events [35], which were not observed in any volunteers. Moreover, blood pressure not only remained within the normal range in all volunteers (with values between 130 and 80 mmHg for SBP and DBP, respectively) [36], but also throughout the study after the OPO and SO dietary intervention (Appendix A). Several animal studies conducted to evaluate the effect of triterpenic acids (specifically oleanolic acid) on endothelial function demonstrated a protective action against blood pressure dysregulation and disorders associated with hypertension [14,37]. However, considering that participants in the present study were not hypertensive, the lack of effect on blood pressure could be due to being normotensive.

Similarly, markers of endothelial function did not show significant changes after the dietary intervention with OPO and SO. Only eNOS was close to reaching the significance level for the effect of oil (*p =* 0.083), with a downward and upward trend after OPO and SO consumption, respectively (Table 6). This enzyme is responsible for the synthesis of nitric oxide (NO), a key element in vasodilation and the maintenance of vascular tone [38]. Therefore, this result reinforces the possible beneficial effect of OPO on endothelial function previously observed in hypertensive animals fed OPO enriched in triterpenic acids (oleanolic acid and maslinic acid) [13,38]. The slight non-significant decrease detected after SO consumption could be related to its high content in linoleic acid, based on a study in endothelial progenitor cells (EPCs) in which incubation with linoleic acid (the main fatty acid in SO) caused a reduction in eNOS expression [39].

### 4.3. Effect of Nutritional Intervention on Inflammation Markers

Given the relationship between cardiovascular risk and inflammatory processes, different biomarkers of inflammation were analysed. According to the linear mixed model applied to the rates of change, OPO resulted in a slight increase in serum levels of the anti-inflammatory cytokines IL-4 (*p* = 0.021) and IL-13 (*p* = 0.023) concentrations, showing a decrease after SO consumption in both normo- and hypercholesterolemic humans (Table 5). IL-4 acts as an anti-inflammatory cytokine by blocking the synthesis of IL-1, TNF-α, IL-6, and macrophage inflammatory protein [40]. It also promotes Th2 lymphocyte differentiation, B-lymphocyte proliferation and differentiation, and is a potent inhibitor of apoptosis [40,41]. The anti-inflammatory cytokine IL-13 modulates the production of IL-1, TNF-alpha, IL-8, and macrophage inflammatory protein. In addition, IL-13 stimulates B cell growth and differentiation, inhibits Th1 cells and the production of inflammatory cytokines [42]. IL-4 and IL-13 have been studied for their involvement in the pathogenesis of allergic disorders such as asthma [40,41] and atopic dermatitis [42] through the activation of eosinophils and the production of immunoglobulin E (IgE) by B cells [43]. Although the assessed inflammation parameters showed significant differences in IL-4 and IL-13, their association with allergic disorders such as asthma suggests that these changes could be driven by factors external to the dietary intervention.

The behaviour of some minor components of OPO, such as squalene [43] or triterpenic acids (ursolic and oleanolic acids) [44,45] on the inflammatory response has already been evaluated in vitro and in animal models. The slight upward trend of IL-4 and IL-13 after OPO intake agrees with that observed in a study by Sánchez-Quesada et al. [43] after treatment with 1 µM squalene of immune cells differentiated into pro-inflammatory M1 macrophages. In contrast, studies carried out in murine models with ursolic acid [45,46] and oleanolic acid [44] have shown a reduction in IL-4 [46] and IL-13 [44,45,46] levels. The relationship between consumption of OPO enriched with triterpenic acids (maslinic and oleanolic acids) and the production of inflammatory mediators has been evaluated in several studies in murine animals [13,14]. In these studies, a lower expression of TNF-α and MCP-1 was found when the animals were fed with OPO enriched with oleanolic and maslinic acids. In the study (Table 5), TNF-α value showed minimal variation after OPO ingestion. However, these results cannot be compared with those obtained in the present trial because they were performed in cultured M1 macrophages [43] or in murine models with asthma [44,45,46], in contrast to our study, where participants did not have a baseline inflammatory state.

The slight decrease observed in IL-4 after SO intake was also observed in a clinical trial conducted in 78 patients after administration of 2.5 g/day of conjugated linoleic acid (CLA), 400 mg/day of vitamin E, or their mixture in adults with rheumatoid arthritis [47]. All other parameters analysed did not reach the level of significance (Table 5). However, in line with Claro-Cala et al. [14], MCP-1 values showed a slight decreasing trend in both groups of volunteers after OPO ingestion without reaching the level of significance. As mentioned above, more nutritional intervention studies with OPO are required to determine if the findings of this study are replicated in other trials.

### 4.4. Effect of Nutritional Intervention on Diabetes and Obesity Markers

Regarding the effect of OPO and SO consumption on parameters associated with insulin resistance and glycaemic homeostasis, Table 7 shows that mean values of glucose, insulin and HbA1c remained within the normal range for the adult population [48]. This outcome was expected as participants did not suffer from type 2 diabetes mellitus and suggests the lack of insulin resistance in the study population. However, significant changes were observed in HbA1c (*p =* 0.021) by group effect (normocholesterolemic and hypercholesterolemic group), and in glucose values by dietary intervention effect (OPO and SO) (*p =* 0.007) (Table 7). According to the Bonferroni test, changes in glucose values (*p =* 0.007) were observed in normocholesterolemic subjects, with a slight increase of 1.3% after OPO intake, and a decrease of 2.6% after SO intake (Table 7). This tendency was contrary to that in hypercholesterolemic subjects, with a slight decrease of 0.3% after OPO intake and an increase of 3.1% after SO intake. However, considering that glucose levels in volunteers of both groups were within the normal range for the adult population (70 and 130 mg/dL) [48], these small changes would likely have minor physiological relevance. The effect of SO on blood glucose levels has been assessed in a recent clinical trial where consumption of 25 mL/day of SO for seven weeks showed no changes in this marker of diabetes [49]. Similarly, a clinical trial conducted with conjugated linoleic acid (3.9 g/day) versus high oleic sunflower oil (3.9 g/day) in 62 healthy subjects for 12 weeks showed no changes in glucose concentrations [50].

The effect of OPO intake on glucose levels has also been evaluated in obese rats after administration of OPO enriched in oleanolic and maslinic acids. The results demonstrated improved oral glucose tolerance compared to obese control mice [14]. Oleanolic acid, which is substantially more abundant in OPO (187 mg/kg) than in SO (<2 mg/kg), has been shown to exert a beneficial effect on glucose metabolism after administration of 55 mL/day of oleanolic acid-enriched olive oil in a clinical trial in 176 pre-diabetic subjects [51]. In addition, animal studies have evaluated the effect of consuming oleanolic acid, either 10 mg/kg daily [52] or 80 mg/kg every three days [53]. The results supported the hypoglycaemic role of this compound by demonstrating a decrease in blood glucose levels [52,53]. Another characteristic component of the triterpene fraction of OPO, ursolic acid, has also shown antidiabetic activity by reducing blood glucose levels and improving glucose tolerance [54]. These results show that OPO rich in triterpenic compounds could positively contribute to glucose metabolism, and more clinical trials in hyperglycaemic volunteers consuming OPO daily are needed to understand the effect of this type of oil.

Finally, the calculation of mathematical equations to measure insulin resistance (HOMA-IR) (*p =* 0.020) and insulin sensitivity (QUICKI index) (*p =* 0.011) showed statistically significant differences between the normocholesterolemic and hypercholesterolemic groups in agreement with the mentioned results of glucose and insulin (Table 7).

Among the obesity-related markers, ghrelin (*p =* 0.031) and leptin (*p =* 0.017), known for their role in appetite regulation, showed significant changes due to the study oils effect (Table 8). The orexigenic hormone ghrelin increased in both groups after dietary intervention with OPO and SO, the increase being more pronounced after SO intake (18.6%) in the risk group. As for the anorexigenic hormone leptin, except for an 8.0% decrease in the normocholesterolemic group after OPO intake, this hormone increased subsequently after intervention with OPO (in the risk group) and SO in both groups of volunteers (Table 8). Studies in obese mice after administration of oleanolic acid (10 mg/kg) and ursolic acid (50 mg/L, in the drinking water) revealed a decrease in ghrelin levels [52,55], contrary to that observed in the present trial after OPO intervention. In relation to leptin, results reported in the literature are contradictory. De Melo et al. [52] and Rao et al. [55] observed an increase in plasma levels of this hormone in mice treated with oleanolic and ursolic acids. In contrast, a study developed by Jia et al. [54] showed decreased levels of leptin after oral administration of ursolic acid (50 and 200 mg/kg), in line with what we observed in healthy volunteers after OPO administration. However, the amount of triterpenic compounds provided by OPO in the present study (equivalent to 8.4 mg/day of oleanolic acid and trace amounts of ursolic acid) is significantly lower than those administered in the aforementioned studies, which would justify the results of the present clinical study.

### 4.5. Effect of Nutritional Intervention on Anthropometric Measurements

One of the most remarkable results associated with OPO and SO consumption was observed in visceral fat (*p =* 0.028), a marker of cardiovascular and metabolic risk. According to Table 9, OPO consumption produced a slight decrease of 4.5% and 1.4% in both groups of volunteers (healthy and at-risk groups), and an increase after SO intake of 2.2% and 6.6% in the normocholesterolemic and hypercholesterolemic groups, respectively. The beneficial effect on anthropometric parameters following consumption of a monounsaturated fat was also observed in an intervention study, where consumption of up to 20 g/day of olive oil for 6 months led to a significant decrease in body weight and BMI [56]. Similarly, administration of oleanolic acid [52] or ursolic acid [55] to obese mice, as well as with OPO enriched in triterpenic acids (oleanolic and maslinic acids) [14], showed not only a decrease in body weight but also a decrease in visceral adiposity. Given that visceral obesity is associated with increased adipocytokine production, pro-inflammatory activity, increased risk of developing diabetes, dyslipidaemia, hypertension, and atherosclerosis [57], this finding supports the idea that OPO consumption could be a nutritional tool to prevent the onset of cardiometabolic diseases. The upward trend observed in the present study following the intake of SO is consistent with the study in which SO and its effect on visceral fat accumulation was evaluated in humans (39 subjects). The study involved overfeeding with PUFA-rich muffins, which showed a lower increase in visceral fat compared with subjects fed saturated fat-rich muffins [58]. Finally, BMI was statistically significantly different between the normocholesterolemic and hypercholesterolemic groups, with higher mean levels in participants with elevated cholesterol levels, supporting the association between altered lipid profile levels (such as high TC levels) and elevated indicators of overweight/obesity. However, no differences were observed due to the consumption of both oils studied (Table 9).

### 4.6. Effect of Nutritional Intervention on Oxidative Status

Three complementary methods such as FRAP, ABTS, and ORAC were applied to determine the antioxidant capacity after nutritional intervention with OPO and SO. According to the linear repeated measures model, consumption of OPO and SO had no effect on any of the parameters used to evaluate the oxidative status of the volunteers (Appendix A). Significant differences in FRAP values were only observed for the effect of belonging to the normocholesterolemic or hypercholesterolemic group (*p =* 0.013). Another marker assessed for oxidative status after OPO and SO consumption was LDLox. Oxidation of this lipoprotein, which plays an important role in the initiation and progression of atherosclerosis, also showed no significant variations throughout the trial (Appendix A). This parameter has been evaluated in different in vitro assays, carried out with different compounds present in OPO, such as erythrodiol, uvaol, oleanolic acid, and maslinic acid, with positive results in the reduction of lipid peroxidation [3]. Furthermore, a diet supplemented with a by-product rich in squalene resulted in lower serum LDLox levels in postmenopausal women [59]. In the present study, a slight decrease of −10.0% (Appendix A) in LDLox concentrations was observed after OPO intervention in at-risk subjects, although without reaching significant differences. These results are in line with those described by Conterno et al. [60] in hypercholesterolemic subjects, where the 8-week consumption of olive pomace powder-enriched cookies tended to reduce LDLox levels, again without reaching the level of significance. These results evidence the interest in conducting further clinical trials with OPO in subjects with high cholesterol levels to confirm the potential effect on LDLox.

On the other hand, the consumption of both vegetable oils (OPO and SO) had no significant effect on serum MDA levels, a widely used marker of lipid peroxidation, in both study groups (Appendix A). There are previous studies demonstrating how triterpenic acids (oleanolic, ursolic, and maslinic acids) [61,62,63,64] and squalene (components of OPO) [65] decrease MDA levels in various animal models of hypertension, inflammation, or diabetes. However, the high, pharmacological dose of pure triterpenic acids and squalene administered to the experimental animals, as well as the pathologies of the murine models used, limit potential comparisons with the present nutritional human intervention study.

Finally, it is important to note that both oils evaluated in this study (OPO and SO) are foods with a high vitamin E content, a nutrient that has been shown to contribute to the protection of cells against oxidative damage (Commission Regulation (EC) No. 432/2012) [66]. Due to the similarity in the vitamin E content in OPO and SO (Table 1), it is possible that no differences were observed in the oxidative status parameters evaluated.

### 4.7. Strengths and Limitations

One of the strengths of this novel clinical trial carried out with OPO was the identification of numerous biomarkers related to cardiovascular disease and its associated pathologies. This approach provides a broad understanding of the role of OPO on health in humans. However, the present study also had limitations on the blinding of volunteers, considering that the colour, smell, and taste of the two oils are different, it was not possible to prevent volunteers (familiar with both oils) from knowing/guessing which oil they were consuming in each stage. Another limitation was that the results reported in the present trial do not allow the generation of clinical evidence because the type I error of 0.05 and statistical power of 80% (1-beta) were taken as a reference in an incidental sample of volunteers controlled only by cholesterol levels. To generate clinical evidence, it would have been necessary to adjust (reduce) alpha and to perform the present study in a population with many other controllable effects (in addition to cholesterol levels).

## 5. Conclusions

Regular consumption of OPO and SO had no statistically significant effect on any of the markers related to lipid profile, blood pressure, and endothelial function in normocholesterolemic and hypercholesterolemic participants. Only eNOS level change was close to being significant due to the effect of oil (OPO and SO) (*p =* 0.083); further studies would be necessary to assess the potential effect of OPO on this biomarker, of great interest considering the involvement of eNOS on the synthesis of nitric oxide, the main factor responsible for vasodilation and maintenance of vascular tone. This result was reinforced by a significant decrease in visceral fat (*p =* 0.028) after OPO intake in both groups, accompanied by the increment of leptin level in the hypercholesterolemic group (*p =* 0.017). These results suggest a potential beneficial effect of sustained consumption of OPO on biomarkers that may have a positive effect on cardiometabolic health. However, more clinical trials on at-risk populations are needed to confirm these health effects.

## Figures and Tables

**Figure 1 nutrients-14-03927-f001:**
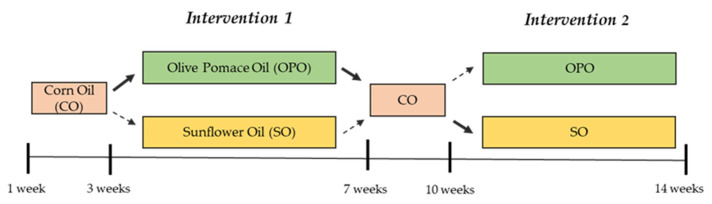
Design of the intervention study with olive pomace oil (OPO) and sunflower oil (SO). Corn oil (CO) was used during run-in and wash-out.

**Figure 2 nutrients-14-03927-f002:**
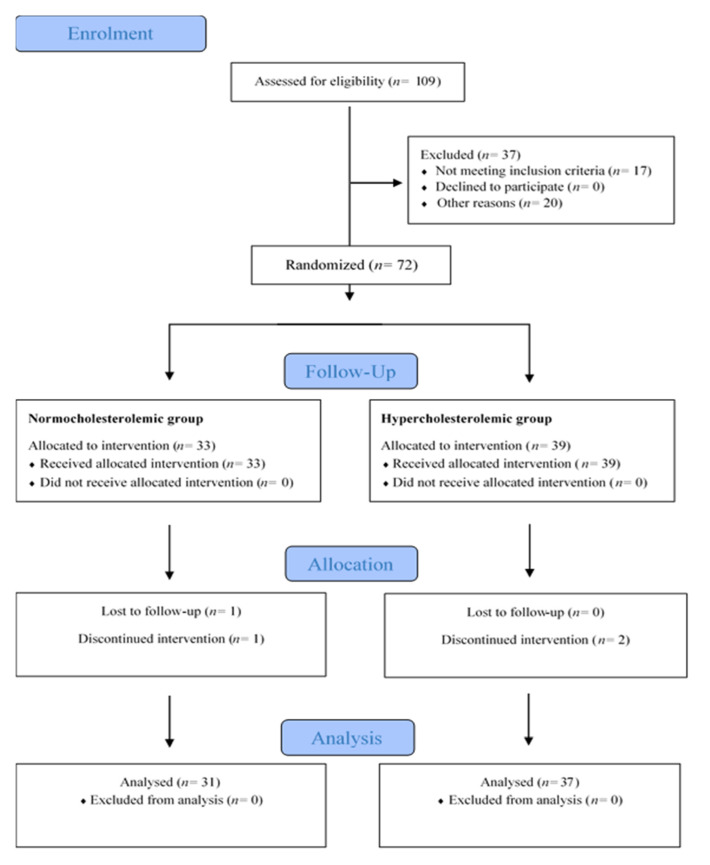
Study flow diagram (Consolidated Standards of Reporting Trials, CONSORT 2010).

**Table 1 nutrients-14-03927-t001:** Chemical composition of olive pomace oil (OPO), sunflower oil (SO), and corn oil (CO).

	OPO	SO	CO
Fatty acids (%)			
C 12:0 (Lauric acid)	-	<0.01	0.01
C 14:0 (Myristic acid)	0.02	0.08	0.04
C 16:0 (Palmitic acid)	10.78	6.43	10.60
C 16:1 (Palmitoleic acid)	0.84	0.14	0.15
C 17:0 (Margaric acid)	0.06	0.03	0.07
C 17:1 (Margaroleic acid)	0.10	0.03	0.05
C 18:0 (Stearic acid)	3.10	3.63	2.03
C 18:1 (Oleic acid)	74.32	29.76	35.36
C 18:2 (Linoleic acid)	9.02	58.46	50.52
C 20:0 (Arachidic acid)	0.48	0.26	0.48
C 18:3 (Linolenic acid)	0.68	0.09	0.04
C 20:1 (Eicosenoic acid)	0.33	0.16	0.31
C 22:0 (Behenic acid)	0.19	0.69	0.17
C 22:1 (Erucic acid)	-	<0.01	<0.01
C 24:0 (Lignoceric acid)	0.08	0.24	0.17
*Trans* Oleic (*t*-C18:1)	0.28	0.03	0.02
*Trans* Linoleic + *Trans* Linolenic (*t*-C18:2 + *t*-C18:3)	0.15	0.19	1.15
Squalene (ppm)			
Squalene (ppm)	675	314	548
Tocopherols (mg/kg)			
α-Tocopherol (Vit. E)	195	217	33
β-Tocopherol	<2	9	<2
γ-Tocopherol	<2	<2	205
δ -Tocopherol	<2	<2	6
Tocotrienols (mg/kg)			
A-Tocotrienol	-	-	10
β-Tocotrienol	-	-	<2
Υ-Tocotrienol	-	-	5
Δ-Tocotrienol	-	-	<2
Sterols (%)			
Cholesterol	0.19	0.12	0.21
Brassicasterol	<0.10	<0.1	0.62
24-Methylcholesterol	<0.10	0.12	0.92
Campesterol	3.06	8.40	20.75
Campestanol	0.16	0.07	0.92
Stigmasterol	1.23	6.90	6.85
Δ7-Campesterol	<0.10	2.70	<0.10
Δ5,23-Stigmastadienol	0.32	0.34	<0.10
Clerosterol	1.08	0.66	0.70
β-Sitosterol	88.64	53.67	61.09
Sitostanol	1.44	0.64	2.11
Δ5-Avenasterol	1.82	3.08	3.90
Δ5,24-Stigmastadienol	1.35	1.08	0.51
Δ7-Stigmastenol	0.50	16.29	0.63
Δ7-Avenasterol	0.19	5.93	0.78
Δ-Sitosterol apparent	94.65	59.47	68.31
Total Sterols (ppm)	3344.2	2820.5	8962.0
Triterpenic alcohols (mg/kg)			
Erythrodiol + Uvaol	745	<1.0	<1.0
Phenols (mg/kg)			
Total phenols	<1.0	<1.0	<1.0
Triterpenic acids (mg/kg)			
Oleanolic acid	187	<2.0	<2.0
Ursolic acid	<2.0	<2.0	<2.0
Maslinic acid	<2.0	<2.0	<2.0
Aliphatic alcohols (mg/kg)			
C22 + C24 + C26 + C28	1681	38	29

**Table 2 nutrients-14-03927-t002:** Baseline characteristics of participants.

	Normocholesterolemic (*n* = 31)	Hypercholesterolemic (*n* = 37)
Men, *n*	8	19
Women, *n*	23	18
Age (years)	30 ± 2	41 ± 2
BMI (kg/m^2^)	23 ± 2	26 ± 1
Waist circumference (cm)	78 ± 2	85 ± 2
Total cholesterol (mg/dL)	172 ± 3	239 ± 5
LDL-cholesterol (mg/dL)	93 ± 3	148 ± 5
Systolic blood pressure (mmHg)	110 ± 2	121 ± 2
Diastolic blood pressure (mmHg)	74 ± 1	81 ± 2

Values represent mean ± SEM. BMI: body mass index; LDL: low-density lipoprotein.

**Table 3 nutrients-14-03927-t003:** Energy intake and dietary components during the intervention trial with the two oils, olive pomace oil (OPO) and sunflower oil (SO) *.

	Normocholesterolemic *n* = 31					*p* Value **	Hypercholesterolemic *n* = 37					*p* Value **
		OPO		SO				OPO		SO		
	Baseline	Initial	Final	Initial	Final	Oil	Baseline	Initial	Final	Initial	Final	Oil
Energy (kcal/day)	1976 ± 70	1906 ± 77	1949 ± 78	1996 ± 85	1982 ± 82	0.314	2081 ± 79	2061 ± 78	2042 ± 72	2025 ± 72	2043 ± 74	0.455
Proteins (g/day)	84 ± 4	78 ± 3	80 ± 3	84 ± 5	81 ± 4	0.621	92 ± 4	88 ± 4	83 ± 4	88 ± 4	81 ± 3	0.103
Carbohydrates (g/day)	187 ± 10	182 ± 12	181 ± 11	186 ± 11	182 ± 10	0.942	195 ± 12	199 ± 10	195 ± 9	188 ± 10	187 ± 10	0.427
Lipids (g/day)	89 ± 4	88 ± 4	90 ± 4	93 ± 4	93 ± 5	0.400	91 ± 4	90 ± 4	93 ± 4	94 ± 4	98 ± 4	0.124
SFA (g/day)	28 ± 2	28 ± 2	28 ± 2	31 ± 2	28 ± 2	0.287	29 ± 2	28 ± 2	28 ± 2	28 ± 2	28 ± 2	0.773
MUFA (g/day)	38 ± 2 ^a^	30 ± 2 ^b^	44 ± 3 ^a^	31 ± 1 ^b^	28 ± 2 ^b^	0.000	41 ± 2 ^a^	30 ± 1 ^b^	45 ± 2 ^a^	31 ± 1 ^b^	31 ± 2 ^b^	0.000
PUFA (g/day)	15 ± 1 ^a^	22 ± 2 ^b^	11 ± 1 ^c^	23 ± 2 ^bd^	29 ± 2 ^d^	0.000	12 ± 1 ^a^	23 ± 1 ^b^	13 ± 1 ^a^	27 ± 1 ^bc^	31 ± 2 ^d^	0.000
Cholesterol (mg/day)	325 ± 23	308 ± 23	337 ± 23	319 ± 19	340 ± 29	0.668	372 ± 21	311 ± 22	317 ± 24	312 ± 21	351 ± 23	0.067
Dietary fibre (g/day)	20 ± 2	17 ± 1	17.5 ± 0.1	18 ± 1	20 ± 1	0.137	22 ± 2	22 ± 1	22 ± 1	22 ± 1	22 ± 1	0.964
Vitamin E (mg/day)	9.7 ± 0.8 ^a^	14 ± 1 ^b^	17 ± 1 ^b^	14 ± 1 ^b^	22 ± 2 ^c^	0.000	120 ± 14 ^a^	121 ± 13 ^b^	137 ± 18 ^c^	146 ± 13 ^c^	114 ± 11 ^d^	0.000

* Values represent mean ± SEM. Data were analysed using a general linear repeated measures model. According to the Bonferroni test, values with different superscript letters correspond to significant differences within the normocholesterolemic N or hypercholesterolemic H groups. *p* values correspond to the effect of taking OPO or SO. Significance level was *p* < 0.05. SFA: saturated fatty acids. MUFA: monounsaturated fatty acids. PUFA: polyunsaturated fatty acids. ** Oil (Baseline vs. OPO vs. SO).

**Table 4 nutrients-14-03927-t004:** Effect of olive pomace oil (OPO) and sunflower oil (SO) consumption on lipid profile and liver function *.

	Normocholesterolemic *n* = 31		Hypercholesterolemic *n* = 37		*p* Value		
	OPO	SO	OPO	SO	Oil	N/H	N/H × Oil
Total-cholesterol (mg/dL)							
Initial	168 ± 5	168 ± 5	221 ± 5	221 ± 6	0.923	0.000	0.868
Final	164 ± 5	169 ± 5	220 ± 5	220 ± 6	0.642	0.000	0.679
Rate of change	−1.4 ± 2.0	1.1 ± 1.8	−0.3 ± 1.6	−0.1 ± 1.6	0.425	0.907	0.813
Triglycerides (mg/dL)							
Initial	73 ± 5	76 ± 6	111 ± 9	111 ± 11	0.778	0.000	0.884
Final	70 ± 5	72 ± 6	109 ± 9	114 ± 11	0.694	0.000	0.975
Rate of change	−2.6 ± 0.04	−1.8 ± 5.0	1.2 ± 4.0	6.9 ± 5.7	0.486	0.215	0.784
HDL-cholesterol (mg/dL)							
Initial	60 ± 2	60 ± 2	65 ± 3	64 ± 3	0.857	0.146	0.814
Final	59 ± 2	60 ± 2	64 ± 2	63 ± 2	0.676	0.179	0.405
Rate of change	−1.2 ± 1.9	0.8± 2.2	0.9 ± 2.0	−2.0 ± 1.9	0.838	0.754	0.342
LDL-cholesterol (mg/dL)							
Initial	93 ± 4	92 ± 4	133 ± 5	134 ± 5	0.919	0.000	0.859
Final	91 ± 4	94 ± 4	134 ± 4	134 ± 5	0.622	0.000	0.801
Rate of change	−0.3 ± 2.8	3.3 ± 2.9	1.1 ± 1.7	1.0 ± 2.2	0.547	0.733	0.657
VLDL-cholesterol (mg/dL)							
Initial	15 ± 1	15 ± 1	24 ± 3	23 ± 3	0.942	0.000	0.713
Final	14 ± 1	14 ± 1	22 ± 2	23 ± 3	0.766	0.000	0.902
Rate of change	−3.0± 4.3	−2.8 ± 4.7	0.2 ± 4.5	5.7 ± 5.8	0.267	0.267	0.783
LDL-cholesterol/HDL-cholesterol							
Initial	1.6 ± 0.1	1.6 ± 0.1	2.2 ± 0.1	2.2 ± 0.1	0.852	0.000	0.823
Final	1.6 ± 0.1	1.6 ± 0.1	2.3 ± 0.1	2.2 ± 0.1	0.788	0.000	0.874
Rate of change	1.7 ± 3.2	4.3 ± 3.7	1.4 ± 2.3	4.4 ± 3.0	0.315	0.945	0.968
Total Cholesterol/HDL-cholesterol							
Initial	2.9 ± 0.1	2.9 ± 0.1	3.6 ± 0.1	3.7 ± 0.2	0.992	0.000	0.990
Final	2.9 ± 0.1	2.9 ± 0.1	3.7 ± 0.2	3.6 ± 0.2	0.778	0.000	0.866
Rate of change	0.26 ± 1.8	1.32 ± 2.2	−0.3 ± 1.8	2.8 ± 2.0	0.942	0.262	0.615
Apolipoprotein (Apo A1) (mg/dL)							
Initial	161 ± 3	163 ± 3	175 ± 4	177 ± 4	0.635	0.000	0.963
Final	165 ± 4	167 ± 3	183 ± 4	179 ± 5	0.397	0.001	0.333
Rate of change	2.98 ± 1.9	3.0 ± 1.7	4.6 ± 1.1	1.0 ± 1.3	0.195	0.849	0.298
Apo B (mg/dL)							
Initial	72 ± 3	73 ± 3	101 ± 3	100 ± 3	0.974	0.000	0.772
Final	77 ± 3	79 ± 3	108 ± 3	108 ± 4	0.833	0.000	0.580
Rate of change	8.8 ± 2.4	9.5 ± 2.3	8.3 ± 1.9	9.0 ± 1.9	0.768	0.846	0.953
Apo B/Apo A1							
Initial	0 46 ± 0.02	0.45 ± 0.02	0.59 ± 0.03	0.56 ± 0.02	0.458	0.000	0.757
Final	0.47 ± 0.02	0.48 ± 0.02	0.61 ± 0.03	0.63 ± 0.03	0.875	0.000	0.970
Rate of change	4.3 ± 3.2	6.3 ± 2.7	5.6 ± 2.4	11.3 ± 2.6	0.126	0.420	0.578
ALAT (UI/L)							
Initial	21 ± 2	21 ± 2	24 ± 3	27 ± 3	0.504	0.052	0.608
Final	19 ± 2	20 ± 2	26 ± 3	28 ± 3	0.498	0.005	0.868
Rate of change	107.7 ± 5.0	109.41 ± 7.5	119.8 ± 5.7	117.0 ± 7.1	0.921	0.119	0.782
ASAT (UI/L)							
Initial	20.6 ± 0.9	21.0 ± 1.2	22.7 ± 1.3	24.1 ± 1.3	0.536	0.033	0.322
Final	18.5 ± 0.8	19.2 ± 0.7	22.5 ± 1.2	24.2 ± 1.3	0.275	0.000	0.635
Rate of change	101.7 ± 3.1	106.9 ± 4.7	113.5 ± 4.6	112.4 ± 4.7	0.559	0.045	0.418

* Values represent mean ± SEM. The table shows the initial (pre-treatment) and final (post-treatment) mean values. The rate of change was calculated from initial and final values as [(final value-initial value)/initial value] and expressed as percentage. Data were analyzed using a linear mixed model. *p* values in the first column correspond to the effect of taking the oil (OPO or SO), those of the penultimate column to the effect of the group [normocholesterolemic (N) or hypercholesterolemic (H)], and the last column to the interaction of oil and group. Significance level was set at *p* < 0.05. Apo: Apolipoprotein. ALAT: Alanine aminotransferase. ASAT: Aspartate aminotransferase.

**Table 5 nutrients-14-03927-t005:** Effect of olive pomace oil (OPO) and sunflower oil (SO) consumption on inflammatory biomarkers *.

	Normocholesterolemic *n* = 31		Hypercholesterolemic *n* = 35 ****		*p* Value		
	OPO	SO	OPO	SO	Oil	N/H	N/H × Oil
CRP (mg/dL)							
Initial	0.23 ± 0.07	0.25 ± 0.10	0.14 ± 0.03	0.34 ± 0.17	0.222	0.823	0.860
Final	0.17 ± 0.05	0.11 ± 0.03	0.17 ± 0.04	0.16 ± 0.04	0.311	0.579	0.438
Rate of change *	39.1 ± 3.1	32.1 ± 0.3	15.1 ± 0.7	17.5 ± 1.1	0.476	0.410	0.815
IL-1β (pg/mL)							
Initial	0.97 ± 0.06	0.95 ± 0.03	0.89 ± 0.02	0.91 ± 0.03	0.785	0.097	0.691
Final	0.92 ± 0.03	0.91 ± 0.03	0.89 ± 0.02	1.05 ± 0.15	0.311	0.980	0.335
Rate of change	−2.4 ± 2.8	−3.2 ± 2.8	4.6 ± 2.3	0.8 ± 14.1	0.449	0.202	0.259
IL-2 (pg/mL)							
Initial	16.4 ± 0.7	16.6 ± 0.5	15.4 ±0.3	15.9 ± 0.5	0.308	0.147	0.766
Final	16.0 ± 0.5	15.6 ± 0.5	15.3 ± 0.4	15.6 ± 0.5	0.573	0.488	0.433
Rate of change	0.1 ± 2.3	−5.2 ± 2.3	−0.4 ± 1.8	−1.4 ± 2.3	0.193	0.544	0.370
IL-4 (pg/mL)							
Initial	2.68 ± 0.06	2.64 ± 0.06	2.57 ± 0.05	2.61 ± 0.05	0.907	0.217	0.430
Final	2.64 ± 0.06	2.59 ± 0.07	2.55 ± 0.04	2.62 ± 0.07	0.515	0.762	0.137
Rate of change	0.7 ± 2.3	−3.0 ± 2.3	2.5 ± 1.6	−1.6 ± 2.1	0.778	0.229	0.021
IL-6 (pg/mL)							
Initial	5.2 ± 0.2	5.4 ± 0.3	5.1 ± 0.2	5.0 ± 0.2	0.947	0.206	0.258
Final	5.3 ± 0.3	5.1 ± 0.3	5.0 ± 0.2	5.2 ± 0.2	0.986	0.571	0.668
Rate of change	3.5 ± 4.4	−2.1 ± 4.4	−3.0 ± 4.3	5.0 ± 3.3	0.816	0.885	0.169
IL-7 (pg/mL)							
Initial	33 ± 1	34 ± 1	33 ± 1	34 ± 1	0.388	0.613	0.782
Final	34 ± 1	34 ± 1	33 ± 1	33 ± 1	0.692	0.610	0.856
Rate of change	2.3 ± 2.5	−1.2 ± 2.5	0.6 ± 1.7	−1.0 ± 2.3	0.252	0.777	0.641
IL-8 (pg/mL)							
Initial	22 ± 2	24 ± 2	20 ± 1	21 ± 1	0.634	0.057	0.861
Final	21 ± 1	21 ± 1	20 ± 1	20 ± 1	0.871	0.633	0.874
Rate of change	−5.6 ± 3.8	−3.2 ± 3.8	3.0 ± 2.5	−3.0 ± 2.2	0.204	0.113	0.568
IL-10 (pg/mL)							
Initial	10.9 ± 0.2	11.1 ± 0.3	10.8 ± 0.3	10.7 ± 0.2	0.319	0.260	0.677
Final	11.0 ± 0.2	10.8 ± 0.2	10.9 ± 0.3	10.6 ± 0.2	0.883	0.552	0.233
Rate of change	1.9 ± 1.6	−2.6 ± 1.6	−0.3 ± 1.1	0.9 ± 1.6	0.310	0.686	0.080
IL-12 (p70) (pg/mL)							
Initial	12.1 ± 0.3	12.4 ± 0.3	11.7 ± 0.3	11.9 ± 0.2	0.287	0.091	0.838
Final	12.2 ± 0.3	11.9 ± 0.3	11.7 ± 0.2	12.0 ± 0.3	0.957	0.565	0.253
Rate of change	1.5 ± 2.5	−4.1 ± 2.5	0.9 ± 1.7	1.0 ± 1.7	0.200	0.293	0.182
IL-13 (pg/mL)							
Initial	3.0 ± 0.1	3.2 ± 0.2	2.8 ± 0.1	2.9 ± 0.1	0.345	0.045	0.992
Final	3.1 ± 0.1	3.0 ± 0.1	2.8 ± 0.1	2.9 ± 0.1	0.826	0.142	0.437
Rate of change	2.8 ± 2.8	−5.2 ± 2.8	1.0 ± 1.9	−0.7 ± 3.0	0.023	0.527	0.159
IL-17 (pg/mL)							
Initial	13.9 ± 0.4	14.0 ± 0.3	13.4 ± 0.3	13.5 ± 0.2	0.475	0.199	0.875
Final	14.0 ± 0.4	13.8 ± 0.4	13.3 ± 0.2	13.5 ± 0.3	0.936	0.143	0.489
Rate of change	0.8 ± 2.1	−0.7 ± 2.1	−0.2 ± 1.6	0.3 ± 1.6	0.773	0.997	0.570
G-CSF (pg/mL)							
Initial	137 ± 3	140 ± 2	134 ± 2	134 ± 2	0.456	0.041	0.453
Final	137 ± 2	137 ± 3	133 ± 2	135 ± 2	0.875	0.150	0.547
Rate of change	0.9 ± 1.7	−1.7 ± 1.7	−0.4 ± 1.0	0.8 ± 1.2	0.618	0.711	0.185
GM-CSF (pg/mL)							
Initial	4.7 ± 0.2	4.7 ± 0.2	4.7 ± 0.3	4.6 ± 0.2	0.726	0.929	0.911
Final	4.7 ± 0.2	4.5 ± 0.2	4.5 ± 0.2	4.8 ± 0.3	0.746	0.772	0.329
Rate of change	0.9 ± 2.6	−2.4 ± 2.6	−1.6 ± 2.3	4.4 ± 3.3	0.651	0.472	0.114
MCP-1 (pg/mL)							
Initial	39 ± 3	37 ± 4	33 ± 2	33 ± 2	0.909	0.088	0.949
Final	35 ± 6	34 ± 3	31 ± 1	32 ± 2	0.574	0.243	0.416
Rate of change	−2.9 ± 3.8	−6.5 ± 3.8	−1.7 ± 3.2	0.3 ± 3.2	0.816	0.273	0.497
MIP-1β (ng/mL)							
Initial	69 ± 6	73 ± 5	71 ± 8	73 ± 8	0.572	0.790	0.793
Final	67 ± 4	66 ± 4	71 ± 8	71 ± 9	0.834	0.453	0.845
Rate of change	4.9 ± 3.9	−8.3 ± 3.9	−1.3 ± 4.1	1.4 ± 6.3	0.361	0.774	0.133
TNF-α (pg/mL)							
Initial	23.1 ± 0.9	23.7 ± 0.8	23.3 ± 1.4	23.9 ± 1.4	0.633	0.897	0.978
Final	23.0 ± 0.8	22.4 ± 0.8	23.3 ± 1.5	23.4 ± 1.5	0.888	0.584	0.644
Rate of change	0.5 ± 1.8	−5.0 ± 1.8	0.0 ± 1.4	−1.5 ± 2.0	0.083	0.567	0.382

* Values represent mean ± SEM. The table shows the initial (pre-treatment) and final (post-treatment) mean values. The rate of change was calculated from initial and final values as [(final value-initial value)/initial value] and expressed as percentage. Data were analyzed using a linear mixed model. *p* values in the first column correspond to the effect of taking the oil (OPO or SO), those of the penultimate column to the effect of the group [normocholesterolemic (N) or hypercholesterolemic (H)], and the last column to the interaction of oil and group. Significance level was set at *p* < 0.05. CRP: C reactive protein. IL: Interleukin. G-CSF: Granulocyte colony-stimulating factor. GM-CSF: Granulocyte–macrophage colony-stimulating factor. IFN-γ: Interferon gamma. MCP-1: Monocyte chemoattractant protein-1. MIP-1β: Macrophage inflammatory protein 1 beta. TNF-α: Tumor necrosis factor alpha. ** Two hypercholesterolemic volunteers were excluded for statistical analysis due to outliers.

**Table 6 nutrients-14-03927-t006:** Effect of olive pomace oil (OPO) and sunflower oil (SO) consumption on endothelial function biomarkers *.

	Normocholesterolemic *n* = 31		Hypercholesterolemic *n* = 37		*p* Value		
	OPO	SO	OPO	SO	Oil	N/H	N/H × Oil
eNOS (ng/mL)							
Initial	0.21 ± 0.03	0.25 ± 0.05	0.22 ± 0.03	0.21 ± 0.03	0.926	0.647	0.597
Final	0.22 ± 0.04	0.20 ± 0.03	0.19 ± 0.03	0.20 ± 0.03	0.944	0.708	0.644
Rate of change	16.03 ± 0.14	−6.67 ± 0.09	15.98 ± 0.14	−4.76 ± 0.12	0.083	0.903	0.912
E-selectin (ng/mL)							
Initial	11 ± 2	15 ± 2	12 ± 2	13 ± 2	0.260	0.939	0.540
Final	11 ± 2	14 ± 3	11 ± 2	10 ± 1	0.588	0.193	0.045
Rate of change	18.0 ± 0.1	14.4 ± 0.1	5.4 ± 0.1	16.5 ± 0.2	0.552	0.342	0.960
P-selectin (ng/mL)							
Initial	214 ± 18	190 ± 12	195 ± 11	195 ± 11	0.412	0.537	0.469
Final	206 ± 15	202 ± 16	179 ± 12	196 ± 11	0.635	0.226	0.448
Rate of change	0.60 ± 0.05	6.86 ± 0.05	−1.72 ± 0.06	2.85 ± 0.04	0.289	0.519	0.871
ICAM-1 (pg/mL)							
Initial	3093 ± 495	3332 ± 502	2954 ± 439	3291 ± 579	0.546	0.854	0.553
Final	2900 ± 452	3302 ± 566	3270 ± 557	3311 ± 643	0.705	0.770	0.601
Rate of change	−0.09 ± 0.05	−1.84 ± 0.04	9.30 ± 0.06	−1.88 ± 0.05	0.120	0.742	0.473
VCAM-1 (pg/mL)							
Initial	11,049 ± 2375	11,961 ± 2422	10,716 ± 2081	12,911 ± 2894	0.426	0.909	0.822
Final	11,280 ± 2523	13,101 ± 3152	12,271 ± 2480	13,917 ± 3574	0.522	0.822	0.621
Rate of change	5.44 ± 0.07	10.51 ± 0.08	19.90 ± 0.08	6.11 ± 0.09	0.373	0.856	0.073

* Values represent mean ± SEM. The table shows the initial (pre-treatment) and final (post-treatment) mean values. The rate of change was calculated from initial and final values as [(final value-initial value)/initial value] and expressed as percentage. Data were analyzed using a linear mixed model. *p* values in the first column correspond to the effect of taking the oil (OPO or SO), those of the penultimate column to the effect of the group [normocholesterolemic (N) or hypercholesterolemic (H)], and the last column to the interaction of oil and group. Significance level was set at *p* < 0.05. eNOS: Endothelial nitric oxide synthase. E-selectin: Endothelial selectin. P-selectin: Platelet selectin. ICAM-1: Intercellular adhesion molecule 1. VCAM-1: Vascular cell adhesion molecule 1.

**Table 7 nutrients-14-03927-t007:** Effect of olive pomace oil (OPO) and sunflower oil (SO) consumption on diabetes markers *.

	Normocholesterolemic *n* = 31		Hypercholesterolemic *n* = 37		*p* Value		
	OPO	SO	OPO	SO	Oil	N/H	N/H × Oil
Glucose (mg/dL)							
Initial	80 ± 1	82 ± 1	83 ± 2	81 ± 1	0.938	0.259	0.130
Final	81 ± 1	79 ± 2	83 ± 1	83 ± 1	0.433	0.403	0.944
Rate of change	1.3 ± 1.2 ^a^	−2.6 ± 1.3 ^b^	−0.3 ± 1.2	3.1 ± 1.3	0.825	0.073	0.007
Insulin (µUI/mL)							
Initial	7.3 ± 0.8	8.3 ± 0.8	10.2 ± 1.5	8.5 ± 0.9	0.977	0.243	0.579
Final	6.2 ± 0.6	6.8 ± 0.7	9.1 ± 0.9	9.1 ± 1.2	0.918	0.007	0.449
Rate of change	−5.6 ± 6.5	−12.5 ± 5.6	4.7 ± 7.6	8.8 ± 5.9	0.077	0.802	0.825
HbA1c (%)							
Initial	5.30 ± 0.05	5.33 ± 0.04	5.33 ± 0.04	5.36 ± 0.04	0.433	0.403	0.944
Final	5.30 ± 0.04	5.29 ± 0.04	5.33 ± 0.03	5.32 ± 0.04	0.837	0.488	0.949
Rate of change	0.1 ± 0. 5	−0.6 ± 0. 6	0.0 ± 0. 4	−0.8 ± 0. 4	0.994	0.021	0.258
HOMA-IR							
Initial	1.4 ± 0.2	1.7 ± 0.2	2.2 ± 0.4	1.7 ± 0.2	0.886	0.186	0.674
Final	1.3 ± 0.1	1.3 ± 0.1	1.9 ± 0.2	1.9 ± 0.3	0.978	0.006	0.560
Rate of change	−3.1 ± 7.4	−14.3 ± 5.8	6.0 ± 8.6	12.4 ± 6.2	0.952	0.020	0.102
HOMA-β							
Initial	174 ± 21	191 ± 27	188 ± 20	140 ± 36	0.503	0.619	0.767
Final	143 ± 14	185 ± 43	186 ± 19	166 ± 15	0.669	0.607	0.185
Rate of change	−8.5 ± 5.3	4.9 ± 5.2	7.1 ± 7.2	−1.7 ± 6.9	0.665	0.633	0.218
QUICKI							
Initial	0.371 ± 0.005	0.363 ± 0.005	0.358 ± 0.006	0.365 ± 0.006	0.931	0.348	0.200
Final	0.379 ± 0.005	0.376 ± 0.006	0.359 ± 0.005	0.363 ± 0.006	0.943	0.005	0.658
Rate of change	2.2 ± 1.3	3.9 ± 1.2	0.7 ± 0.9	−0.7 ± 1.1	0.891	0.011	0.096

* Values represent mean ± SEM. The table shows the initial (pre-treatment) and final (post-treatment) mean values. The rate of change was calculated from initial and final values as [(final value-initial value)/initial value] and expressed as percentage. Data were analyzed using a linear mixed model. *p* values in the first column correspond to the effect of taking the oil (OPO or SO), those of the penultimate column correspond to the effect of the group [normocholesterolemic (N) or hypercholesterolemic (H)], and the last column to the interaction of oil and group. Significance level was set at *p* < 0.05. HbA1c: haemoglobin A1c. HOMA-IR: Homeostatic model to assessment insulin resistance. HOMA-β: Homeostatic model to assess β-cell functionality. QUICKI: Quantitative insulin sensitivity check index.

**Table 8 nutrients-14-03927-t008:** Effect of olive pomace oil (OPO) and sunflower oil (SO) consumption on diabetes and obesity biomarkers *.

	Normocholesterolemic *n* = 31		Hypercholesterolemic *n* = 37		*p* Value		
	OPO	SO	OPO	SO	Oil	N/H	N/H × Oil
C-peptide (pg/mL)							
Initial	548 ± 45	546 ± 43	687 ± 63	634 ± 43	0.582	0.030	0.788
Final	489 ± 33	504 ± 38	639 ± 46	630 ± 43	0.937	0.001	0.873
Rate of change	−6.2 ± 4.7	−3.3 ± 4.7	1.9 ± 5.8	1.8 ± 3.7	0.697	0.259	0.723
Ghrelin (pg/mL)							
Initial	319 ± 42	297 ± 26	271 ± 26	252 ± 23	0.642	0.159	0.971
Final	299 ± 38	309 ± 30	250 ± 23	296 ± 26	0.161	0.330	0.543
Rate of change	3.8 ± 7.9	6.0 ± 5.8	2.3 ± 7.4	41.2 ± 20.7	0.031	0.348	0.166
GIP (pg/mL)							
Initial	807 ± 168	840 ± 188	667 ± 29	635 ± 36	0.639	0.281	0.549
Final	770 ± 162	837 ± 188	657 ± 24	647 ± 38	0.847	0.581	0.571
Rate of change	−1.9 ± 5.7	0.2 ± 2.9	0.4 ± 3.0	3.1 ± 2.9	0.554	0.513	0.975
GLP-1 (pg/mL)							
Initial	255 ± 12	252 ± 13	259 ± 9	251 ± 10	0.624	0.852	0.656
Final	261 ± 11	257 ± 10	256 ± 9	256 ± 10	0.842	0.775	0.896
Rate of change	−3.1 ± 7.7	−4.4 ± 7.8	0.1 ± 2.8	4.0 ± 3.4	0.992	0.902	0.274
Glucagon (pg/mL)							
Initial	865 ± 71	807 ± 64	833 ± 54	858 ± 62	0.806	0.871	0.512
Final	906 ±74	824 ± 71	903 ± 55	869 ± 53	0.351	0.751	0.679
Rate of change	16.8 ± 14.7	5.5 ± 7.1	14.54 ± 6.1	7.97 ± 5.6	0.956	0.480	0.627
Insulin (pg/mL)							
Initial	176 ± 25	200 ± 23	246 ± 33	211 ± 24	0.739	0.114	0.198
Final	150 ± 16	162 ± 20	235 ± 32	234 ± 32	0.841	0.005	0.818
Rate of change	−0.6 ± 9.1	−9.3 ± 9.2	12.0 ± 10.3	15.0 ± 8.1	0.886	0.065	0.807
Leptin (ng/mL)							
Initial	3.3 ± 0.4	3.2 ± 0.4	3.6 ± 0.5	3.0 ± 0.4	0.535	0.940	0.645
Final	2.9 ± 0.4	3.2 ± 0.4	3.2 ± 0.4	3.4 ± 0.4	0.593	0.519	0.911
Rate of change	−8.0± 5.4	1.7 ± 6.3	8.6 ± 6.8	18.6 ± 98.3	0.017	0.180	0.947
PAI-1 (ng/mL)							
Initial	6.9 ± 0.3	7.0 ± 0.3	7.2 ± 0.5	6.9 ± 0.4	0.815	0.853	0.608
Final	7.1 ± 0.3	6.9 ± 0.3	7.1 ± 0.4	7.2 ± 0.4	0.847	0.565	0.590
Rate of change	4.5 ± 3.3	−1.5 ± 2.8	5.7 ± 6.3	9.3 ± 8.1	0.805	0.459	0.444
Resistin (ng/mL)							
Initial	6.0 ± 0.6	6.6 ± 0.7	6.5 ± 0.6	7.2 ± 1.0	0.179	0.641	0.377
Final	6.1 ± 0.4	6.4 ± 0.6	6.1 ± 0.6	6.3 ± 0.5	0.744	0.969	0.858
Rate of change	22.1 ± 0.217.5	8.1 ± 10.4	5.5 ± 11.7	11.7 ± 24.7	0.826	0.221	0.719
Visfatin (ng/mL)							
Initial	1.53 ± 0.09	1.36 ± 0.08	1.63 ± 0.06	1.56 ± 0.07	0.100	0.097	0.596
Final	1.44 ± 0.07	1.48 ± 0.16	1.62 ± 0.07	1.62 ± 0.07	0.764	0.050	0.971
Rate of change	−5.3 ± 2.2	3.9 ± 20.6	0.5 ± 3.0	5.7 ± 4.2	0.724	0.091	0.470
Adiponectin (ng/mL)							
Initial	39 ± 8	56 ± 12	71 ± 17	78 ± 21	0.157	0.264	0.461
Final	49 ± 9	61 ± 20	91 ± 29	63 ± 15	0.029	0.793	0.916
Rate of change	20.4 ± 33.1	63.1 ± 28.2	54.4 ± 28.8	−69.8 ± 36.9	0.205	0.081	0.098
Adipsin (pg/mL)							
Initial	660 ± 68	677 ± 107	853 ± 92	815 ± 91	0.866	0.072	0.739
Final	619 ± 73	600 ± 65	716 ± 64	664 ± 61	0.523	0.190	0.619
Rate of change	4.6 ± 13.2	9.6 ± 11.1	−6.0 ± 1.0	−6.5 ± 9.1	0.969	0.260	0.621

* Values represent mean ± SEM. The table shows the initial (pre-treatment) and final (post-treatment) mean values. The rate of change was calculated from initial and final values as [(final value-initial value)/initial value] and expressed as percentage. Data were analyzed using a linear mixed model. *p* values in the first column correspond to the effect of taking the oil (OPO or SO), those of the penultimate column correspond to the effect of the group [normocholesterolemic (N) or hypercholesterolemic (H)], and the last column to the interaction of oil and group. Significance level was set at *p* < 0.05. GIP: Gastric inhibitory polypeptide. GLP-1: Glucagon like peptide 1. PAI-1: Plasminogen activator inhibitor-1.

**Table 9 nutrients-14-03927-t009:** Effect of olive pomace oil (OPO) and sunflower oil (SO) consumption on anthropometric measurements and body composition *.

	Normocholesterolemic *n* = 31		Hypercholesterolemic *n* = 37		*p* Value		
	OPO	SO	OPO	SO	Oil	N/H	N/H × Oil
Weight (kg)							
Initial	64.1 ± 2.1	64.0 ± 2.1	74.1 ± 2.6	74.1 ± 2.7	0.990	0.000	0.967
Final	63.9 ± 2.1	64.0 ± 2.1	74.2 ± 2.7	74.2 ± 2.7	0.945	0.000	0.967
Rate of change	−0.3± 0.3	0.0 ± 0.2	0.1 ± 0.2	0.1 ± 0.2	0.660	0.272	0.545
BMI (kg/m^2^)							
Initial	23.9 ± 0.7	23.7 ± 0.7	26.0 ± 0.7	26.0 ± 0.7	0.893	0.002	0.976
Final	23.8 ± 0.7	23.8 ± 0.7	26.1 ± 0.7	26.1 ± 0.7	0.930	0.001	0.980
Rate of change	−0.4 ± 0. 3	0.3 ± 0.5	0.2 ± 0.2	0.4 ± 0.3	0.365	0.050	0.805
Body fat (%)							
Initial	25.5 ± 1.5	24.8 ± 1.5	26.2 ± 0.7	26.0 ± 0.7	0.803	0.512	0.845
Final	25.9 ± 1.3	25.2 ± 1.5	26.1 ± 0.7	25.8 ± 0.7	0.844	0.805	0.818
Rate of change	1.1 ± 2.3	−1.0 ± 1.1	3.3 ± 2.4	3.8 ± 2.2	0.848	0.058	0.714
Visceral fat (%)							
Initial	3.5 ± 0.5	3.3 ± 0.5	7.3 ± 0.7	7.0 ± 0.7	0.794	0.000	0.935
Final	3.4 ± 0.5	3.4 ± 0.5	7.2 ± 0.7	7.1 ± 0.7	0.822	0.000	0.867
Rate of change	−4.5 ± 2.7	2.2 ± 4.0	−1.4 ± 1.5	6.6 ± 6.1	0.028	0.490	0.636
Waist circumference (cm)							
Initial	75.8 ± 1.8	76.6 ± 2.2	84.5 ± 2.3	85.6 ± 2.5	0.545	0.000	0.863
Final	74.7 ± 1.8	76.0 ± 2.0	83.7 ± 2.3	85.1 ± 2.3	0.548	0.000	0.997
Rate of change	−1.2 ± 0. 9	−0.7 ± 0. 9	−1.0± 0. 4	−0.5 ± 0. 8	0.467	0.935	0.884
Hip circumference (cm)							
Initial	96.1 ± 1.5	96.0 ± 1.7	99.7 ± 1.2	98.8 ± 1.6	0.973	0.053	0.886
Final	97.6 ± 1.4	96.7 ± 1.7	100.2 ± 1.3	99.2 ± 1.6	0.848	0.001	0.948
Rate of change	0.6 ± 0.5	0.6 ± 0. 9	0.6 ± 0. 4	1.1 ± 0.9	0.534	0.616	0.538
Arm circumference (cm)							
Initial	28.5 ± 0.6	28.3 ± 0.6	30.5 ± 0.6	30.4 ± 0.6	0.848	0.001	0.948
Final	28.3 ± 0.6	28.6 ± 0.7	30.3 ± 0.6	30.5 ± 0.6	0.945	0.002	0.920
Rate of change	−0.5 ± 0. 5	0.1 ± 0. 4	−0.7 ± 0. 5	0.2 ± 0. 4	0.130	0.962	0.559

* Values represent mean ± SEM. The table shows the initial (pre-treatment) and final (post-treatment) mean values. The rate of change was calculated from initial and final values as [(final value-initial value)/initial value] and expressed as percentage. Data were analyzed using a linear mixed model. *p* values in the first column correspond to the effect of taking the oil (OPO or SO), those of the penultimate column correspond to the effect of the group [normocholesterolemic (N) or hypercholesterolemic (H)], and the last column to the interaction of oil and group. Significance level was set at *p* < 0.05. BMI: Body mass index.

## Data Availability

The data presented in this study are available on request from the corresponding author. The data are not publicly available due to privacy concerns.

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
