# Peer review of "Effect of Olive Pomace Oil on Cardiovascular Health and Associated Pathologies"

_nutrients, 2022, doi:10.3390/nu14193927_

Round 1
Reviewer 1 Report (Previous Reviewer 1)
The study by Gonzalez-Ramila et al analyzes the possible beneficial role of
Olive pomace oil (OPO) on different aspects. It is a blinded clinical trial. The study is well conducted. However, there are some issues that must be addressed.
1. Why do the authors study two different kind of subjects? The aim of the study is “to determine the possible beneficial role of OPO on 61 biomarkers of cardiovascular health and associated pathologies (hypertension, inflamma-62 tion, diabetes and obesity)”. No mention is made in the manuscript of the "utility" of studying these two groups.
The sample size is calculated without taking these two groups into account. Why weren't both groups studied together?
2. Why 109 individuals were initially included? It is a “strange” number.
3. The authors say that it is a long duration study. However, 4 weeks is not too long. They must explain why they say this.
4. The conclusions are not appropriate.
- “Regular consumption of OPO and SO had no statistically significant effect on any of 709 the markers related to lipid profile, blood pressure and endothelial function in normocholesterolemic and hypercholesterolemic participants, except for eNOS levels, which changes were close to be significant due to the effect of oil (OPO and SO) (p=0.083)”. They cannot say except because there is not statistically significant difference.
- “These results would suggest a possible beneficial effect of OPO on cardiovascular health”. However, only one of the parameters shows beneficial changes. This conclusion should be qualified
5. How do the authors explain that many parameters behave differently in each of the groups (normocholesterolemia and hypercholesterolemia)?
6. I consider that the statistical analysis is not well carried out.
- N/H oil column is confusing. The authors must explain what is “N/H oil”.
- The authors talk about the difference between subjects with and without hypercholesterolemia. However, I do not really know how it was done. What has been compared? The increase, the basal levels, ...?
- Have the authors evaluated the differences between the increments?
- Authors should modify the way of expressing statistical significance. A column should be clearly included for each group of individuals, and not jointly
Author Response
The authors would like to thank the editor and referees for their revision. We feel that their comments have greatly contributed to improving our work. The changes made in the manuscript have been shaded in yellow (referee 1), blue (referee 2) and green (referee 3).
The study by Gonzalez-Ramila et al analyzes the possible beneficial role of olive pomace oil (OPO) on different aspects. It is a blinded clinical trial. The study is well conducted. However, there are some issues that must be addressed.
Point 1: Why do the authors study two different kind of subjects? The aim of the study is “to determine the possible beneficial role of OPO on biomarkers of cardiovascular health and associated pathologies (hypertension, inflamma-tion, diabetes and obesity)”. No mention is made in the manuscript of the "utility" of studying these two groups.
Response 1: Thank you for your comment. Due to previous literature linking the different OPO components, including oleic acid, with cardioprotective effects, the main primary outcome was the potential changes in lipid levels. For this reason, the target population was subjects at high cardiovascular risk (moderately hypercholesterolemic volunteers). However, given the lack of previous human studies performed with olive pomace oil (OPO), we thought it would be of interest to include another population group representing the normal consumer; therefore, the effect of OPO on healthy volunteers was also addressed. In this way, we could know its potential health properties both in a normal population and in subjects at risk of cardiovascular disease. In response to your comment, we have improved the sentence in order to explain the purpose of studying these two groups (shaded in yellow) (Lines 64-65; revised manuscript).
The sample size is calculated without taking these two groups into account. Why weren't both groups studied together?
Response: Thank you for your comment. Total cholesterol was the principal variable taken to calculate the sample size in the population group (described in Sample size calculation and statistical analysis section). Since one group was formed by healthy subjects and the other by hypercholesterolemics, the changes in total cholesterol could be different. Therefore, it was best to calculate the size for each group and the study was carried out in both together.
Point 2: Why 109 individuals were initially included? It is a “strange” number.
Response 2: 109 individuals were the number of people who answered our initial calls during recruitment showing interest in the study and that were recorded in the database and interviewed to assess eligibility. Of these, 72 met the inclusion criteria and were included in the study, although only 68 completed the intervention.
Point 3: The authors say that it is a long duration study. However, 4 weeks is not too long. They must explain why they say this.
Response 3: When we say that it is a long study, we refer to the total extension of the clinical trial, 14 weeks. Indeed, 4 weeks was the time volunteers consumed each study oil, which is not a very long time. Nevertheless, the duration of each intervention was based on previous crossover and controlled clinical trials carried out with virgin and extra virgin olive oils, which lasted between 3 and 4 weeks. [1-7].
[1] Maki, K.C.; Lawless, A.L.; Kelley, K.M.; Kaden, V.N.; Geiger, C.J.; Palacios, O.M.; Dicklin, M.R. Corn oil intake favorably impacts lipoprotein cholesterol, apolipoprotein and lipoprotein particle levels compared with extra-virgin olive oil. Eur. J. Clin. Nutr. 2017, 71, 33-38.
[2] Castañer, O.; Fitó, M.; López-Sabater, M.C.; Poulsen, H.E.; Nyyssönen, K.; Schröder, H.; Salonen, J.T.; De la Torre-Carbot, K.; Zunft, H.-F.; De la Torre, R.; Bäumler, H.; Gaddi, A.V.; Saez, G.T.; Tomás, M.; Covas, M.-I.; EUROLIVE Study group. The effect of olive oil polyphenols on antibodies against oxidized LDL. A Randomized Clinical Trial. Clin. Nutr. 2011, 30, 490-493.
[3] Cherki, M., Derouiche, A., Drissi, A., El Messal, M., Bamou, Y., Idrissi-Ouadghiri, A., Khalil, A., & Adlouni, A. (2005). Consumption of argan oil may have an antiatherogenic effect by improving paraoxonase activities and antioxidant status: Intervention study in healthy men. Nutr. Metab. Cardiovasc. Dis. 2005, 15, 352–360.
[4] Visioli, F.; Caruso, D.; Grande, S.; Bosisio, R.; Villa, M.; Galli, G.; Sirtori, C.; Galli, C. Virgin Olive Oil Study (VOLOS): vasoprotective potential of extra virgin olive oil in mildly dyslipidemic patients. Eur. J. Nutr. 2005, 44, 121-127.
[5] Covas, M.-I.; Nyyssönen, K.; Poulsen, H.E.; Kaikkonen, J.; Zunft, H.-J.F.; Kiesewetter, H.; Gaddi, A.; de la Torre, R.; Mursu, J.; Bäumler, H.; et al. The effect of polyphenols in olive oil on heart disease risk factors. Ann. Intern. Med. 2006, 145, 333–341.
[6] Sánchez-Rodríguez, E.; Biel-Glesson, S.; Fernandez-Navarro, J.R. et al. Effects of virgin olive oils differing in their bioactive compound contents on metabolic syndrome and endothelial functional risk biomarkers in healthy adults: A randomized double-blind controlled trial. Nutrients 2018, 10, 626.
[7] Sánchez-Rodríguez, E.; Lima-Cabello, E.; Biel-Glesson, S. et al (2019) Effects of virgin olive oils differing in their bioactive compound contents on biomarkers of oxidative stress and inflammation in healthy adults: a randomized double-blind controlled trial. Nutrients 2019, 11, 561.
Point 4: The conclusions are not appropriate.
“Regular consumption of OPO and SO had no statistically significant effect on any of 709 the markers related to lipid profile, blood pressure and endothelial function in normocholesterolemic and hypercholesterolemic participants, except for eNOS levels, which changes were close to be significant due to the effect of oil (OPO and SO) (p=0.083)”. They cannot say except because there is not statistically significant difference.
“These results would suggest a possible beneficial effect of OPO on cardiovascular health”. However, only one of the parameters shows beneficial changes. This conclusion should be qualified
Response 4: In response to your comment, the conclusion has been qualified (Line 716-720; revised manuscript).
Point 5: How do the authors explain that many parameters behave differently in each of the groups (normocholesterolemia and hypercholesterolemia)?
Response 5: Given that hypercholesterolemia is not an independent risk factor, but is linked to other pathologies such as obesity, diabetes or hypertension [8-10], it is possible that moderately elevated cholesterol levels in the hypercholesterolemic group are accompanied by other slightly above-average markers, such as blood pressure or anthropometric measurements, without reaching risk values. Therefore, the differences between groups when comparing the initial mean values and the final mean values were expected, and changes occurred preferentially in the group with the most altered values. Similarly, in previous studies carried out in our group, we have observed a different response in normocholesterolemic and hypercholesterolemic subjects, being bigger the response in the latter group, in agreement with the present study.
[8] Ramón-Arbués, E.; Martínez-Abadía, B.; Gracia-Tabuenca, T.; Yuste-Gran, C.; Pellicer-García, B.; Juárez-Vela, R.; Guerrero-Portillo, S.; Sáez-Guinoa, M. (2019). Prevalence of overweight/obesity and its association with diabetes, hypertension, dyslipidemia and metabolic syndrome: a cross-sectional study of a sample of workers in Aragón, Spain. Nutr. Hosp. 2019, 36, 51–59.
[9] Klop, B.; Elte, J.W.; Cabezas, M.C. Dyslipidemia in obesity: mechanisms and potential targets. Nutrients 2013, 5, 1218-1240.
[10] Xi, Y.; Niu, L.; Cao, N.; Bao, H.; Xu, X.; Zhu, H.; Yan, T.; Zhang, N.; Qiao, L.; Han, K.; Hang, G.; Wang, W.; Zhang X. Prevalence of dyslipidemia and associated risk factors among adults aged ≥35 years in northern China: a cross-sectional study. BMC Public. Health. 2020, 20, 1068.
Point 6: I consider that the statistical analysis is not well carried out.
N/H oil column is confusing. The authors must explain what is “N/H oil”.
Response 6: N/H*oil is the interaction of the oil treatment with the group (N/H) (indicated in the footnotes of the table). The purpose of this analysis was to test whether treatment with OPO and SO followed the same direction in the healthy and at-risk groups. When this analysis (N/H*oil) was significant, there was a different response. Thus, the Bonferroni test was applied to find out where the differences occurred.
The authors talk about the difference between subjects with and without hypercholesterolemia. However, I do not really know how it was done. What has been compared? The increase, the basal levels, ...?
Response 6: The tables show the mean initial values (pre-treatment), mean final values (post-treatment) and the rate of change, that is, relative changes from initial values. In our manuscript, we refer mainly to rates of change because it allows us to know the direction of a variable after dietary intervention. The initial and final values are shown to provide further information. The p values indicate the following:
- Oil treatment (OPO vs SO): Effect of OPO and SO intake (in both groups) in the initial phase (pre-treatment), in the final phase (post-treatment) and on the rate of change.
- N/H (N vs H): Behaviour of the two groups (N/H) was compared at initial stage, final stage and rate of change for each variable (TC, LDL, etc.). In case of initial and final stage, we compared whether the mean values before (initial stage) and after (final stage) treatment (with OPO and SO) were different between groups. For the rates of change, we compared the relative changes from initial values between the two groups.
- N/H*Oil (N*OPO vs H*OPO; N*SO vs H*SO): In this case, we analysed the interaction between the oil treatment and the group, i.e. the effect of consuming OPO and SO in the normocholesterolemic and the hypercholesterolemic groups.
Have the authors evaluated the differences between the increments?
Response 6: Yes, the rates of change allow knowing the direction (upward or downward) of each variable. The statistical method applied in the present study (linear mixed model) allowed to compare the direction (rate of change) of each variable after OPO and SO treatment (independent of the group) (oil effect), the direction in the normocholesterolemic group compared to the hypercholesterolemic group (group effect), and whether the direction was the same in the N and H group after OPO and SO intake (N/H*Oil).
Authors should modify the way of expressing statistical significance. A column should be clearly included for each group of individuals, and not jointly
Response 7: Thank you for your comment. If we segment the data and include a p value for each group, it would allow us to know the behaviour of OPO and SO in healthy and at-risk subjects, but not to compare each other. In contrast, with the general statistical model applied (linear mixed model) and with the fixed and random factors included for analysis, multiple comparisons are obtained that allow us to know the behaviour of prolonged consumption of the oils studied in both normocholesterolemic and hypercholesterolemic people. We, therefore, consider that the statistical analysis of the data is adequated.

Reviewer 2 Report (New Reviewer)
The manuscript entitled “Effect of olive pomace oil on cardiovascular health and associated pathologies” falls within the scope of the Journal. However, this reviewer has the following comments on the manuscript:
-I suggest that Supplementary Table 2 be included as a regular table in the manuscript.
- Authors should report the keywords in alphabetical order.
- Authors should insert an abbreviation section. The words for which are specified an abbreviation should be written in full the first time they are mentioned.
- Authors should include more recent literature and insert further adequate references in the Introduction and Results section in order to support their study.
- Material and methods section is detailed, but it needs more literature references.
- Authors should insert a graphical abstract that summarizes the contents of the article in a concise form in order to capture the attention of the readership.
- The English language has to be extensively revised.
- Authors should improve the formal aspects of the manuscript.
Author Response
The authors would like to thank the editor and referees for their revision. We feel that their comments have greatly contributed to improving our work. The changes made in the manuscript have been shaded in yellow (referee 1), blue (referee 2) and green (referee 3).
The manuscript entitled “Effect of olive pomace oil on cardiovascular health and associated pathologies” falls within the scope of the Journal. However, this reviewer has the following comments on the manuscript:
Point 1: I suggest that Supplementary Table 2 be included as a regular table in the manuscript.
Response 1: Thank you for your suggestion. The Table has been inserted in the manuscript (shaded in blue) (Lines 400-411; revised manuscript).
Point 2: Authors should report the keywords in alphabetical order.
Response 2: Thank you for your appreciation. The keywords have been reordered (Lines 25-26; revised manuscript).
Point 3: Authors should insert an abbreviation section. The words for which are specified an abbreviation should be written in full the first time they are mentioned.
Response 3: We have revised all abbreviated words to confirm that they are defined the first time they are mentioned in each section (abstract, main text and tables/figures). In addition, we have collected all abbreviations and incorporated them at the end of the text (Lines 770-788; revised manuscript).
Point 4: Authors should include more recent literature and insert further adequate references in the Introduction and Results section in order to support their study.
Response 4: For the development of the manuscript, we have carried out an extensive bibliographic review of the studies developed with the oils used in the present clinical trial. Therefore, we consider that all references related to the study are cited, including the recent.
Point 5: Material and methods section is detailed, but it needs more literature references.
Response 5: The material and methods section has been reviewed. To the best of our knowledge, all methods are properly cited. Nevertheless, we are open to consider any suggestion.
Point 6: Authors should insert a graphical abstract that summarizes the contents of the article in a concise form in order to capture the attention of the readership
Response 6: Following your appreciation, a graphical abstract has been prepared. A screenshot is attached below.
Point 7: The English language has to be extensively revised.
Response 7: According to your comment, all authors (s) of the manuscript (one of them is native) have revised the manuscript in order to improve the applied language.
Point 8: Authors should improve the formal aspects of the manuscript.
Response 8: Thank you for your comments. The authors have carefully revised the manuscript according to the guidelines provided by the journal.

Reviewer 3 Report (New Reviewer)
Dear authors,
your effort is intersting aiming to contribute on the possible dietary importance of an oil generally neglected so far although it contains except for a-tocopherol a substantial amount of triterpenoids, which are bioactive.
A few comments are the following:
There is no iformation about the origin of the samples, no discussion how representative is their composition comparared to literature values or considering the standards of international olive council for pomace, and/or the codex alimentarius STANDARD FOR NAMED VEGETABLE OILS CXS 210-1999, EC Reg 2568/91. where the oils fresh and before or close to the best before date? the oils were characterized for their commposition but not for qualiy characteristics such as peroxide value, which is important. Other quality criteria are incuded in the aforementioned sources
the consummmption dosage of 45g/d used should be justified or explained? Is it equal to maximum quantity suggested by national dietary guides for oil consumption?
The authors have already published an almost analogous work (Foods 2022, 11(15), 2186; https://doi.org/10.3390/foods11152186) before almost a month using OPO vs HOSO and SO, which is not included in the present list of references. Thus, 1) the novelty of the present study when supported does not take into account that work 2) consequently the findings are not discussed, at least regarding the common parameters measured in both studies.
l.49 minority compounds use minor components
l.50 the presence of phenolic commpounds in OPO is stated, as well as in other parts of the manuscript. To my knowledge this is not expected due to refining process and i.e. the authors measuring phenols did not find any. A question also i have regarding the tocopherols as e.g. in US standard it is stated that "Alpha-tocopherol is permitted to restore natural tocopherol lost in the refining process for refined olive pomace and olive-pomace oil. Maximum level: 200 mg/kg of total alpha-tocopherol is permitted in the final product".
The selection of CO for the washout period is not clearly understood
In Table 1 "gamma" is missing from tocopherol
in l. 509-513 it is stated that . In addition, oleic acid, as the main fatty acid in all categories of olive oil (EVOO, AOV, AO and OPO), allows these oils to be classified as monounsaturated fats with the potential to reduce coronary heart disease risk, as had already been recognized by the FDA (Food and Drug Administration 2004)
I would like to highlight that in the (EU) No 432/2012 of 16 May 2012 establishing a list of permitted health claims made on foods, other than those referring to the reduction of disease risk and to children's development and health
there is also a health claim for oleic acid/mono-unsaturates, as well as polyunsaturates under specific conditions
Author Response
The authors would like to thank the editor and referees for their revision. We feel that their comments have greatly contributed to improving our work. The changes made in the manuscript have been shaded in yellow (referee 1), blue (referee 2) and green (referee 3).
Dear authors,
your effort is interesting aiming to contribute on the possible dietary importance of an oil generally neglected so far although it contains except for a-tocopherol a substantial amount of triterpenoids, which are bioactive.
A few comments are the following:
Point 1: There is no information about the origin of the samples, no discussion how representative is their composition compared to literature values or considering the standards of international olive council for pomace, and/or the codex alimentarius STANDARD FOR NAMED VEGETABLE OILS CXS 210-1999, EC Reg 2568/91. where the oils fresh and before or close to the best before date? the oils were characterized for their commposition but not for qualiy characteristics such as peroxide value, which is important. Other quality criteria are incuded in the aforementioned sources.
Response 1: Thank you for your comment. ACESUR, which is a leading group in Spain’s national olive oil sector and certainly complies with the highest quality standard regulation (https://acesur.com/en/quality-and-food-safety) following Regulation (EEC) No. 2568/91, provided the oils used in this intervention study. These were oils commercialized for consumption, only the company sent them to us in blind bottles as specified in the manuscript. Therefore, they complied with all the regulations for vegetable oils.
The oils were provided by the collaborating company freshly prepared and bottled, and we distributed them to the volunteers only a few days later, when the clinical trial started, for immediate consumption. The oils that were not provided to the volunteers during the different stages were properly stored under dry, cold, dark conditions to ensure they maintained their properties. All were consumed within 6 months of receiving the oils. This, along with the fact that the best-before date for consumption of the intervention oils (OPO, SO and CO) was two years after packaging, allow us to ensure that their physic-chemical quality and properties were adequately maintained during the study.
Since our objective was to determine the health properties of the OPO's minor components, we focused exclusively on the characterization of these components, assuming that the received oils met all the quality parameters for each oil category. However, recent reports on quality parameters (peroxide value, acidity, polar compounds, etc.) of olive pomace oils similar to the one used in the present study have been recently published by Holgado et al. (2021) and Ruiz-Mended et al (2021), showing an excellent quality of OPO in comparison with other widely used oils such as sunflower or high-oleic acid sunflower oils.
Holgado, F.; Ruiz-Méndez, M.V.; Velasco, J.; Márquez-Ruiz, G. Performance of Olive-Pomace Oils in Discontinuous and Continuous Frying. Comparative Behavior with Sunflower Oils and High-Oleic Sunflower Oils. Foods 2021, 10, 3081.
Ruiz-Méndez, M.-V.; Márquez-Ruiz, G.; Holgado, F.; Velasco, J. Stability of Bioactive Compounds in Olive-Pomace Oil at Frying Temperature and incorporation into Fried Foods. Foods 2021, 10, 2906.
Finally, according to your comment, the characterization of the oils has been discussed in the manuscript (shaded in green) (Line 510-523; revised manuscript).
Point 2: the consumption dosage of 45g/d used should be justified or explained? Is it equal to maximum quantity suggested by national dietary guides for oil consumption?
Response 2: The justification for the use of 45 g/day of OPO, SO and CO is explained in the manuscript as follows: “All volunteers consumed 45 g/day of oil according to the intervention phase (OPO, SO and CO) to cover 20% of the daily energy intake of monounsaturated fats (equivalent to 44-67 g/d for 2000-3000 Kcal/d, respectively) following the Spanish Society of Community Nutrition (SENC) recommendations” (Line: 118-121; revised manuscript).
Point 3: The authors have already published an almost analogous work (Foods 2022, 11(15), 2186; https://doi.org/10.3390/foods11152186) before almost a month using OPO vs HOSO and SO, which is not included in the present list of references. Thus, 1) the novelty of the present study when supported does not take into account that work 2) consequently the findings are not discussed, at least regarding the common parameters measured in both studies.
Response 3: Thank you for your appreciation. Our research group has developed two clinical intervention studies comparing OPO with high oleic sunflower oil (first trial, under review) and SO (present study). The article you mention (Foods) studies the comparison of the two clinical trials, being a continuation of this clinical trial. Therefore, the novelty of this work would not be affected. Although the idea was to publish the present paper before Foods article, the publication process has been slowed down for reasons beyond our control. For all these reasons, and due to the proximity in time of these two works, we do not consider it necessary to refer to Foods' work in the discussion.
Point 4: l.49 minority compounds use minor components.
Response 4: The phrase has been rewritten.
Point 5: l.50 the presence of phenolic commpounds in OPO is stated, as well as in other parts of the manuscript. To my knowledge this is not expected due to refining process and i.e. the authors measuring phenols did not find any. A question also i have regarding the tocopherols as e.g. in US standard it is stated that "Alpha-tocopherol is permitted to restore natural tocopherol lost in the refining process for refined olive pomace and olive-pomace oil. Maximum level: 200 mg/kg of total alpha-tocopherol is permitted in the final product".
Response 5: In the final step of the OPO production, up to 20% virgin olive oil or extra virgin olive oil is added to the refined pomace oil to provide it with organoleptic and nutritional characteristics (a process known in Spain as “encabezado”) (REGULATION (EC) No 1513/2001). For this reason, phenolic compounds can be detected in the characterization of the OPOs. The line you mention (line 50) refers to a scientific publication by Mateos et al. (2020) that reports that the content of phenolic compounds in OPO is usually lower than 100 mg/kg, as has been described by other authors. However, the content of phenolic compounds in our oils was almost non-existent, so the discussion has not claimed that OPO's beneficial effects are due to these components.
Regarding the tocopherol content, as mentioned above the oils used in the present clinical trial were supplied by an external company (ACESUR), and we do not know the precise process used to obtain these oils. As just mentioned, a small proportion of virgin or extra virgin olive oil is usually added in the “encabezado” process, which might also contribute to the final tocopherol content of OPO. However, we aimed to characterize the study oils to know their chemical composition regardless of the process used to obtain them.
Point 6: The selection of CO for the washout period is not clearly understood.
Response 6: For the run in/wash out period, we selected a seed oil with fatty acid and minor compound composition different from that of OPO and the control oil (sunflower oil). For this reason, we selected a seed oil such as corn oil (CO) with a mild flavour. To clarify this point, we have added a sentence in lines 66-67 (revised manuscript).
Point 7: In Table 1 "gamma" is missing from tocopherol.
Response 7: Thank you for your comment, the mistake has been rectified in the table.
Point 8: in l. 509-513 it is stated that . In addition, oleic acid, as the main fatty acid in all categories of olive oil (EVOO, AOV, AO and OPO), allows these oils to be classified as monounsaturated fats with the potential to reduce coronary heart disease risk, as had already been recognized by the FDA (Food and Drug Administration 2004).
I would like to highlight that in the (EU) No 432/2012 of 16 May 2012 establishing a list of permitted health claims made on foods, other than those referring to the reduction of disease risk and to children's development and health.
there is also a health claim for oleic acid/mono-unsaturates, as well as polyunsaturates under specific conditions
Response 8: Thank you for your input. In line with your comment, we have included in the discussion section the authorized health claims for vitamin E, contained in Commission Regulation (EU) 432/2012 (Oxidative and antioxidant biomarker analysis section) (section 4.6) (Line 728-732; revised manuscript).
On the other hand, the claim of oleic acid and monounsaturated fats under Commission Regulation (EU) 432/2012, on the effects on lipid profile related to the substitution of saturated fats by unsaturated fats (such as MUFA or PUFA) in the diet has not been included in the manuscript. The reason is that in our study, volunteers replaced their usual oil (mainly virgin olive oil and, in some cases, SO) with 45 g/d of mainly monounsaturated (OPO) and polyunsaturated (SO) fats.

Round 2
Reviewer 3 Report (New Reviewer)
The authors addressed most of the issues I raised, so I would recommend the publication, although i would insitist at least in incorporating their other publication as reference in the introduction, as there it is expexted to cover what has been done on the topic, which is usefull for the reader
Author Response
The authors would like to thank the editor and referees for their revision. We feel that their comments have greatly contributed to improving our work. The changes made in the manuscript have been shaded in green.

This manuscript is a resubmission of an earlier submission. The following is a list of the peer review reports and author responses from that submission.
Round 1
Reviewer 1 Report
The study by Gonzalez-Ramila et al analyzes the possible beneficial role of olive pomace oil (OPO) on different aspects. It is a blinded clinical trial. The study is well conducted. However, there are some issues that must be addressed.
1. Why do the authors study two different kind of subjects? The aim of the study is “to determine the possible beneficial role of OPO on 61 biomarkers of cardiovascular health and associated pathologies (hypertension, inflamma-62 tion, diabetes and obesity)”. No mention is made in the manuscript of the "utility" of studying these two groups.
The sample size is calculated without taking these two groups into account. Why weren't both groups studied together?
2. Why 109 individuals were initially included? It is a “strange” number.
3. The authors say that it is a long duration study. However, 4 weeks is not too long. They must explain why they say this.
4. The conclusions are not appropriate.
- “Regular consumption of OPO and SO had no statistically significant effect on any of 709 the markers related to lipid profile, blood pressure and endothelial function in normocholesterolemic and hypercholesterolemic participants, except for eNOS levels, which changes were close to be significant due to the effect of oil (OPO and SO) (p=0.083)”. They cannot say except because there is not statistically significant difference.
- “These results would suggest a possible beneficial effect of OPO on cardiovascular health”. However, only one of the parameters shows beneficial changes. This conclusion should be qualified
5. How do the authors explain that many parameters behave differently in each of the groups (normocholesterolemia and hypercholesterolemia)?
6. I consider that the statistical analysis is not well carried out.
- N/H oil column is confusing. The authors must explain what is “N/H oil”.
- The authors talk about the difference between subjects with and without hypercholesterolemia. However, I do not really know how it was done. What has been compared? The increase, the basal levels, ...?
- Have the authors evaluated the differences between the increments?
- Authors should modify the way of expressing statistical significance. A column should be clearly included for each group of individuals, and not jointly
Reviewer 2 Report
This study has too low number of participants ( n=68) and is of too short duration ( 4 weeks) to prove the hypothesis that pomace oil can have an effect on cardiovascular health and associated pathologies
Moreover there are no clear inclusion criteria listed